# Using the linear references from the pangenome to discover missing autism variants

Yang Sui [1], Jiadong Lin[1], Michelle D. Noyes[1], Youngjun Kwon [1], Isaac Wong [1], Nidhi Koundinya [1], William T. Harvey [1], Mei Wu[1], Kendra Hoekzema[1], Katherine M. Munson [1], Gage H. Garcia [1], Jordan Knuth [1], Julie Wertz [1], Tianyun Wang [2,3,4], Kelsey Hennick [5,6], Druha Karunakaran[7], Rafael A. Polo Prieto [8,9], Rebecca Meyer-Schuman[8,9], Fisher Cherry [8,9], Davut Pehlivan[8,9,10,11], Bernhard Suter[10,11], Jonas A. Gustafson [12,13], Danny E. Miller [12,14], Human Pangenome Reference Consortium (HPRC)*, Hanna Berk-Rauch[7], Tomasz J. Nowakowski [5,6], Aravinda Chakravarti [7,15], Huda Y. Zoghbi [8,9,16,17,18,19] & Evan E. Eichler [1,20] ✉

To better understand large-effect pathogenic variation associated with autism, we generated long-read sequencing (LRS) data to construct phased and near-complete genome assemblies (average contig N50 = 43 Mbp, QV = 56) for 189 individuals from 51 families with unsolved cases. We applied read- and assembly-based strategies to facilitate comprehensive characterization of de novo mutations, structural variants (SVs), and DNA methylation. Using LRS pangenome controls, we efficiently filtered >97% of common SVs exclusive to 87 offspring. We find no evidence of increased autosomal SV burden for probands when compared to unaffected siblings yet observe a suggestive trend toward an increased SV burden on the X chromosome among affected females. We establish a workflow to prioritize potential pathogenic variants by integrating autism risk genes and putative noncoding regulatory elements defined from ATAC-seq and CUT&Tag data from the developing cortex. In total, we identified three pathogenic variants in *TBL1XR1*, *MECP2*, and *SYNGAP1*, as well as nine candidate de novo and biallelic inherited homozygous SVs, most of which were missed by short-read sequencing. Our work highlights the potential of phased genomes to discover complex more pathogenic mutations and the power of the pangenome to restrict the focus on an increasingly smaller number of SVs for clinical evaluation.

Autism is a class of neurodevelopmental disorders (NDDs) characterized by challenges in social interaction, communication, and repetitive behaviors, with symptoms and severity varying widely among individuals. Currently, the median prevalence of autism globally is approximately 1%, with a male-to-female ratio of about 4 to 1[1]. Genetic investigations into rare variants have typically focused on gene-disruptive de novo mutations (DNMs), rare inherited variants, and copy number variants (CNVs) discovered through meta-analyses of short-read sequencing (SRS) and earlier array CGH. These variants account for an estimated 20% of autism cases and have led to the

A full list of affiliations appears at the end of the paper. *A list of authors and their affiliations appears at the end of the paper. ✉e-mail: ee3@uw.edu

discovery of ~1000 risk genes and numerous CNVs associated with NDDs[2–9]. The underlying mutations for the remaining cases, including structural variants (SVs) mapping to repetitive regions, remain poorly understood.

SVs, including deletions (DEL), insertions (INS), inversions (INV), translocations, large-scale CNVs, and other complex rearrangements, are defined as affecting ≥50 base pairs of DNA. They have been shown to have larger effects[10,11] because they can disrupt coding or noncoding regulatory regions, alongside protein-coding genes, thereby playing a critical role in gene regulation and human disease[12–14]. However, many SVs occur in technically and methodologically challenging regions, particularly repetitive sequences, making them difficult to detect and completely characterize using conventional SRS approaches associated with whole-genome sequencing (WGS) or whole-exome sequencing (WES). Similarly, a subset of smaller single-nucleotide variants (SNVs) and indels can be missed by SRS because of their association with low-complexity and repetitive DNA[2,5–7].

Long-read sequencing (LRS) data (15-30 kbp) significantly enhances the sensitivity of variant detection, especially in repetitive DNA regions[15]. Recent studies have revealed that LRS data provide access to ~91% of the human genome, substantially increasing DNM discovery by ~30% and SV discovery by over 47% when compared to SRS datasets[14,16–21]. Consequently, LRS has been increasingly applied to a variety of unsolved patients and disorders to enhance pathogenic variant discovery, although most studies to date have involved relatively modest cohorts focused almost entirely on read-based discovery[22–27]. For example, Hiatt et al. reanalyzed 10 NDD families and 86 probands using Pacific Biosciences (PacBio) high-fidelity (HiFi) LRS and found an additional yield of 7.3% beyond SRS, mainly in the coding regions[22,23]. Sanchis-Juan et al. applied Oxford Nanopore Technologies (ONT) LRS to complement SRS in four probands[24]. Moreover, the ability to accurately call methylated CpGs, especially from ONT LRS, has the added benefit of simultaneously discovering potential imprinting defects[28].

Beyond read-based variant discovery, the combination of LRS technologies (ONT and PacBio) has facilitated the construction of near-complete telomere-to-telomere (T2T) genome assemblies as part of the Human Genome Structural Variation Consortium (HGSVC) and the Human Pangenome Reference Consortium (HPRC)[18,29,30]. These consortia recently made hundreds of diverse human genomes publicly available. This resource is potentially valuable to the clinical genetics community because variant discovery is more complete, providing a control to assess the frequency of variants in regions typically unassayable by SRS and therefore absent or unreliable in associated databases such as gnomAD[31,32]. Moreover, assembly-based comparisons between offspring and parental genomes have been shown to further increase the power to discover DNM by essentially eliminating reference biases[33]. Notwithstanding, the number of samples still remains modest, and much larger sample sizes will be required to understand the full spectrum, especially for those with lower minor allele frequency (MAF).

Using LRS assembly approaches, we sought to construct reference-free genome assemblies comparable to the HGSVC and HPRC controls for all members of autism families. In this study, we present our initial LRS and assembly resource of 189 individuals from 51 unsolved autism families where no pathogenic variant was previously identified in the proband via conventional Illumina whole-genome, whole-exome, or gene panel testing with SRS (Methods). To build reference resources, we constructed near-complete genomes for each individual and assigned a workflow to systematically identify variants from high-quality assemblies; we then compared them with HPRC and HGSVC population controls to discover and validate de novo and rare variants (<0.5%) as further candidates for autism. Importantly, constructing genomes comparable to pangenome references allowed us to dramatically reduce the variant search space for

SVs, highlighting the increasing utility of the pangenome for disease variant discovery.

## Results

### Sequence and assembly of genomes from unsolved autism families

We focused on the sequence and assembly of genomes from 51 unsolved simplex autism families. The set included 46 families (174 individuals) with idiopathic autism from the Simons Simplex Collection (SSC) and Study of Autism Genetics Exploration (SAGE) and five families with a diagnosis of Rett syndrome (15 individuals) (Fig. 1a, Supplementary Data 1). The Rett families had been previously screened using either gene panels and/or WES with no *MECP2* pathogenic mutation reported by clinical testing labs after multiple attempts (Supplementary Data 1, Methods). Similarly, the 46 families were part of large-scale CNV and WGS initiatives over the last decade, where no pathogenic variant had been reported by multiple groups, including our own[2,4,34,35]. For 36 families, there was also an unaffected sibling serving as a genetic control (17 of these quads were sex-matched).

We developed an LRS workflow to enhance variant discovery using reference-level quality genomes with a particular emphasis on characterizing previously undetected SVs and DNMs. First, we sequenced all 189 genomes using PacBio HiFi sequencing technology from peripheral blood ($n = 139$), cell lines ($n = 16$), or a mixture of both ($n = 34$) when DNA from peripheral lymphocytes was limited. Per sample, we generated an average of 36-fold sequence coverage with an overall N50 read length of 19 kbp after extensive quality control (Fig. 1b, Supplementary Data 1, Methods, Supplementary Fig. 1). Using parental Illumina reads and hifiasm[36], we generated haplotype-resolved genome assemblies (i.e., phased by parent-of-origin) for each of the 87 offspring. The resulting assemblies are highly contiguous (average contig N50 of 43 Mbp (Fig. 1c and Supplementary Fig. 2)) and highly accurate (quality value (QV) = 56). The assembly quality is comparable to that of control pangenomes (with mean QV of 56) from the HPRC[30] and HGSVC[18,29] (Supplementary Fig. 2).

### Variant discovery

For each family, SNVs and small indel callsets were generated by the GATK[37] and DeepVariant[38] callers using T2T-CHM13v2.0-aligned HiFi reads mainly from blood samples ($n = 172$). Putative DNMs were further validated by confirming their presence in ONT (Supplementary Fig. 3) and Illumina data, as previously described[33,39]. We discovered on average 96 DNMs per child ($n = 78$), with approximately 83% predicted to be germline and the remainder arising postzygotically. SVs in each individual were identified from the phased assemblies using PAV[18] with GRCh38 as the reference genome. GRCh38 was selected due to the broad availability of annotation resources mapped to this assembly, many of which were integrated into our analyses to facilitate comprehensive functional characterization of putative disruptive variants. Each SV was considered validated if supported by at least one of the alignment-based SV callers, either PacBio structural variant caller (PBSV), Sniffles[40], or both (Methods). We aggregated the validated SVs from all 189 study samples and compared them with SVs observed in the 108 population controls from the HPRC and HGSVC[18,29,30] via Truvari[41], focusing on rare SVs exclusive to autism families. As a result of subsequent LRS of additional 1000 Genomes Project (1KGP) samples recently released as part of the HPRC and other efforts[42], we expanded our SV callset from controls in a staged manner, allowing us to access variants of reduced MAF (see section "Pangenome increased sensitivity for pathogenic SV discovery", Supplementary Fig. 12).

Based on PAV analysis of each child's assembly, we identified an average of 27,576 SVs per diploid genome of which 20,716 SVs (95% Confidence Intervals = 9) were also supported by alignment-based methods (Fig. 2a). We note that two Rett-like samples where DNA was limited (HYZ204_p1 and HYZ207_p1) have lower contig N50 and lower

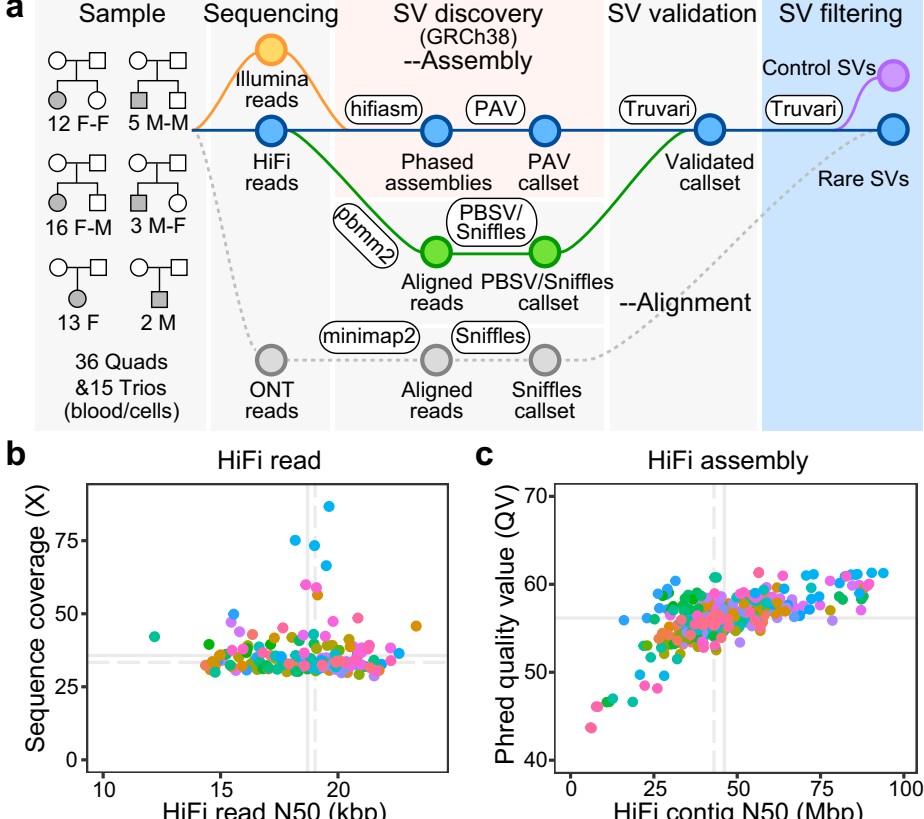

**Fig. 1 | Long-read sequencing and assembly. a** Schematic workflow of LRS data generation and SV discovery with pedigree structures of the 51 unsolved autism families (F = female; M = male). LRS data (PacBio HiFi and ONT) and phased genomes were constructed using hifiasm[36]; SVs were discovered by PAV and validated via Truvari[41] with read-based callers, PBSV and Sniffles (analyses tools indicated in oval boxes). Validated SVs were filtered using a pangenome of 108 control genomes from the HPRC and HGSVC to define a rare SV callset private to the autism families (Supplementary Data 1). **b** HiFi reads N50 and genomic coverage per sample (members of the same family are color coded). **c** Sequence accuracy (QV) and contig N50 length for each HiFi-phased genome assembly. Solid lines represent mean values, while dashed lines indicate median values. Source data are provided in Supplementary Data 1.

assembly QV, resulting in fewer validated SVs (Fig. 2a, Supplementary Fig. 4). The average Mendelian concordance rate for SVs across 87 trios is 90.4% with many of the discrepant alleles associated with multiallelic variants, such as variable number tandem repeats (VNTRs) where both sequence and alignment artefacts confound variant calling[43]. This high-confidence SV callset, thus, affects 11,074,300 bp (95% Confidence Intervals = 4273) per sample (0.4% of the genome). We applied the same SV discovery approach to 108 pangenome controls resulting in on average 24,341 ± 36 validated SVs per diploid genome (Supplementary Fig. 4). This approach improved the sensitivity of control SV detection and facilitated more effective filtering of SVs in the sample set. Both pangenome controls and study samples are from diverse superpopulations. And, as expected, genomically diverse African samples had higher numbers of SVs than non-Africans (Supplementary Fig. 4). We integrated the two callsets for a total of 271,375 nonredundant SVs (Supplementary Data 2) and observed an expected SV size distribution with modes at 300 bp and 6 kbp, corresponding to Alu and LINE retrotransposition events, respectively (Supplementary Fig. 5).

After filtering with the pangenome we identified a total of 33,548 nonredundant SVs (57,716 genotyped SVs, Supplementary Data 3) that were exclusively observed in the 87 children (51 probands and 36 unaffected siblings). At the individual sample level, we effectively filtered ~97% of SVs per child, resulting in approximately 663 rare SVs per child for further consideration (affected vs. unaffected, Z = −0.23, p = 0.82, two-sided Mann-Whitney U Test, Fig. 2a). After testing for Mendelian inheritance, the autism set was further reduced to 25,272

nonredundant SVs. We then classified rare SVs into six categories (Fig. 2b), including autosomal heterozygous (n = 33,701), autosomal homozygous (n = 6639), X chromosome heterozygous (n = 1129), X chromosome homozygous (n = 277), and hemizygous events from males on the X (n = 565) and Y (n = 141) chromosomes. We evaluated the remaining low-confidence SV calls through a suite of tools that test for transmission and de novo variants (Methods). We validated a total of 36 de novo SVs in 51 probands (n = 21) and 36 siblings (n = 15) that were absent from 108 controls (affected vs. unaffected, p = 1, $\chi^2$ test).

## SV burden analyses

Because the majority of rare SVs (96%) map to noncoding DNA, we annotated all SVs for regulatory potential as well as association with known NDD genes. To define putative regulatory elements (REG), we integrated published datasets from ENCODE with regulatory sequences predicted from single-cell and bulk ATAC-seq and CUT&Tag experiments performed on developmentally staged material (16–24 weeks) from the cerebral cortex[44] (Methods). The additional annotation allowed us to identify another 6171 SVs associated with brain-derived regulatory regions (brainREG) beyond those defined by ENCODE intersection and consequently a 45% (6171/13,773) increase in potential SVs affecting noncoding regulatory DNA (Fig. 2c, d). For NDD candidate genes, we primarily focused on those previously reported[5–7,45]. We considered three classes of SVs, namely: de novo, homozygous, and private inherited SVs defined as those observed only once in the parental population[2]. On autosomes, we identified 9474

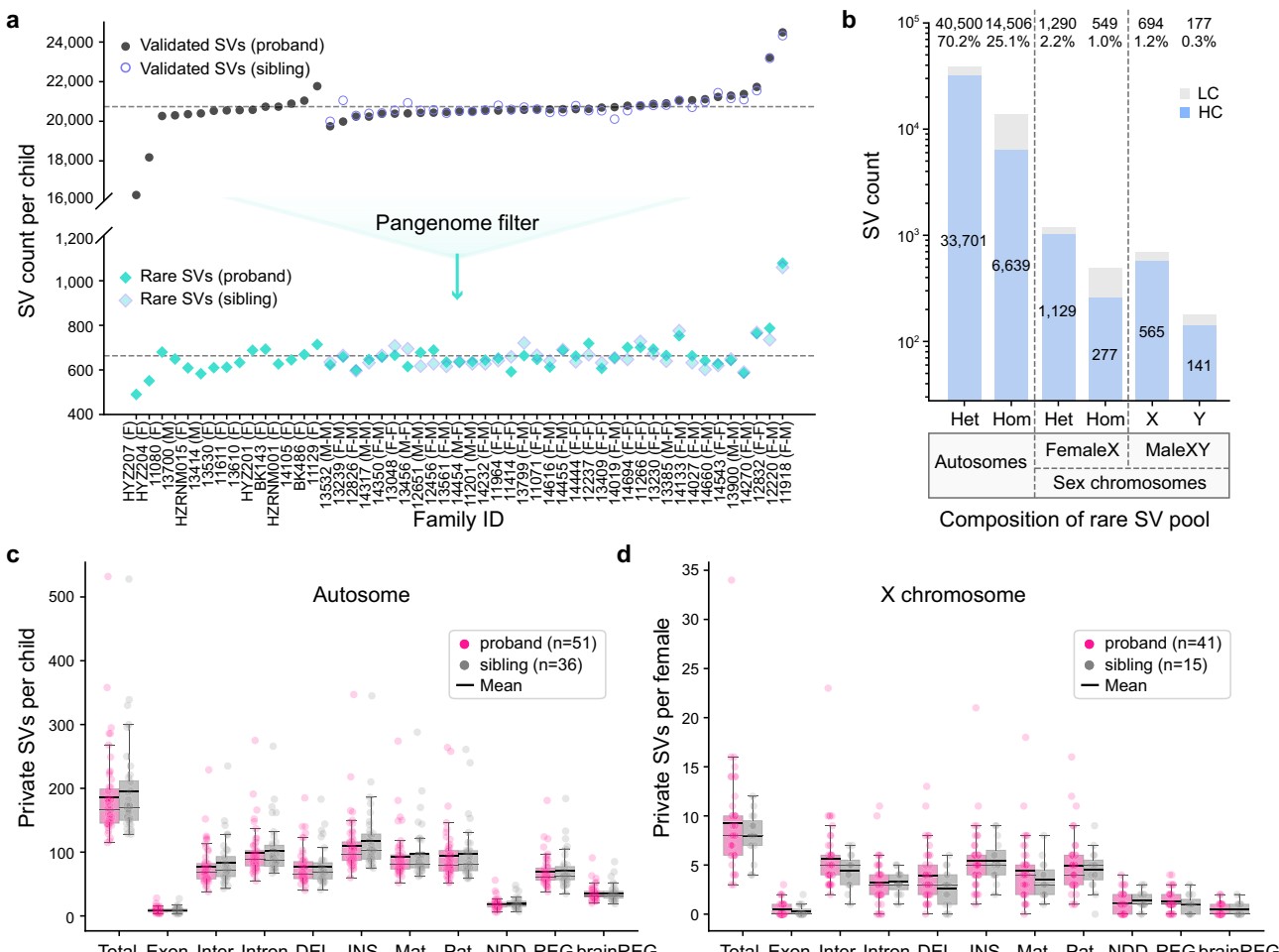

**Fig. 2 | SV discovery, filtering and burden in autism families. a** SV discovery in probands (dark color) and unaffected siblings (light color with blue border) before (top) and after (bottom) pangenome filtering for 51 families with idiopathic autism. Proband sex versus unaffected sibling is shown in parentheses after family IDs. **b** High- (HC) and low-confidence (LC) SVs by genotype class for autosomes and sex chromosomes. Het: heterozygous SVs. Hom: homozygous SVs. HC: high-confidence SVs confirmed by Mendelian inheritance of parental SV calls. LC: low-confidence SVs that initially deviated from Mendelian inheritance patterns in the collapsed table but were subsequently curated through further evaluation. The box plots compare the autosomal private SV burden (**c**) and X chromosome burden (**d**, females only) between probands (pink) and unaffected siblings (gray) for 51 probands (41 females and 10 males) and 36 unaffected siblings (15 females and 21 males). Different functional categories of SV classes are considered: protein-coding and UTR (Exon), intergenic (Inter), intronic (Intron), deletions (DEL), insertions (INS), paternally inherited (Pat), maternally inherited (Mat), those overlapping neurodevelopmental disorder (NDD) genes, brain-derived regulatory regions (brainREG)[44], and a combined set of all regulatory regions (REG). The center lines of the box plots represent the median; box limits indicate upper and lower quartiles; whiskers show 1.5x the interquartile range; individual points represent private SV counts per sample. The black line indicates the mean private count per sample. No significant differences ($\chi^2$ test p-values exceeding 0.05) in the number of SVs between probands and siblings were observed across these categories, with a still insignificant trend observed on the X chromosome for enrichment of SVs on affected females compared to unaffected sisters. Source data are provided as a Source Data file.

private SVs in 51 probands and 7034 private SVs in 36 unaffected siblings. The rates of autosomal SVs (Fig. 2c, Supplementary Fig. 6) do not differ significantly between probands and unaffected siblings across various categories (nominal $p > 0.05$, OR < 1, $\chi^2$ test).

For X chromosome SVs, we analyzed males and females separately. Female probands exhibited a slight excess of X chromosome private SVs compared to the unaffected female siblings (nominal $p = 0.29$, OR = 1.16, adjusted $p = 1$, $\chi^2$ test), particularly for deletions (Fig. 2d, nominal $p = 0.09$, OR = 1.49, adjusted $p = 1$, $\chi^2$ test). We also considered 1417 private homozygous SVs that were inherited from two heterozygous parents on autosomes in only one family but were never observed as homozygous in any controls. Once again, we observed a slight enrichment in the number of these private homozygous SVs in probands relative to unaffected siblings (Supplementary Fig. 7, $p > 0.05$, $\chi^2$ test).

## Sex chromosome assembly analyses

In addition to SVs, we constructed nearly complete X and Y chromosomes by leveraging LRS and parental sequence data. Excluding the pseudoautosomal region (PAR), centromere, and highly repetitive Yq12 heterochromatic regions, we estimate that on average 95% of the X chromosome and 70% of the Y chromosome (Fig. 3a, b, Supplementary Fig. 8) can be aligned (Methods). The assemblies served two purposes: they validated DNMs on the sex chromosomes and allowed parental transmission to be fully assessed (i.e., paternal Y and maternal vs. paternal X chromosomes) without the use of a reference genome (Fig. 3c, d).

The analysis highlighted three hemizygous tandem repeat (TR) expansion outliers in male probands (Supplementary Fig. 9, Methods), each inherited from the mother in the intron of *IL1RAPL1* (Interleukin 1 Receptor Accessory Protein Like 1, SFARI score 2 gene), the intron of *F9*

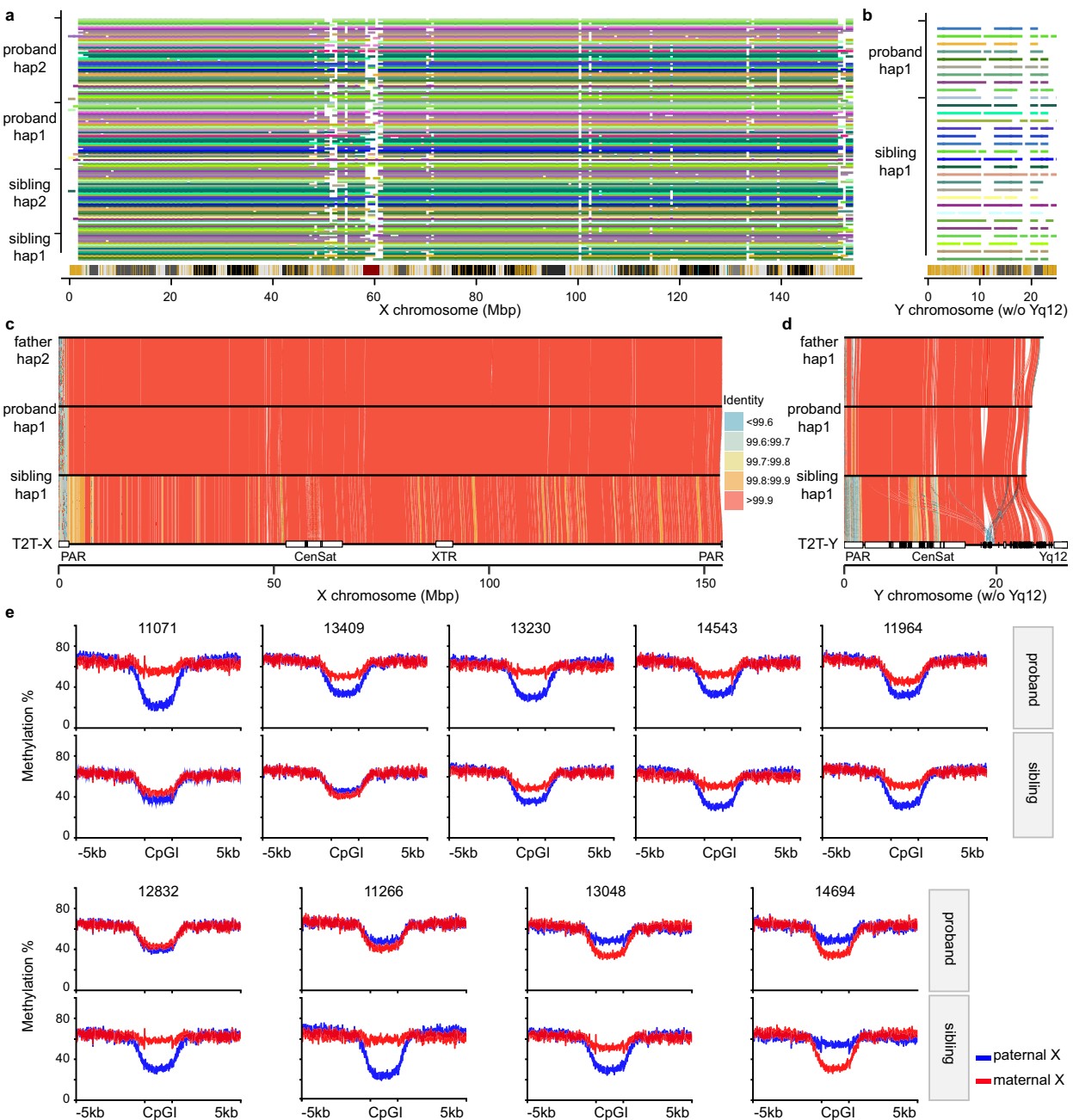

**Fig. 3 | Sex chromosome assembly, transmission and X chromosome inactivation skewing. a** Stacked barplot showing X chromosome assembly continuity and mappability relative to the T2T-CHM13v2.0 reference across haplotypes. Each horizontal line represents one haplotype. The assembled contigs in each haplotype traverse the 1 Mbp window of the reference (no more than 3) and have at least ≥95% sequence overlap. Colored segments indicate SDs (yellow), centromeres (red), and gaps (black) on the reference cytogenetic band. **b** Continuity and mappability of Y chromosome assemblies relative to the T2T-CHM13v2.0 reference (Yq12 heterochromatic region was masked). **c** Transmitted X assemblies from father to two daughters in the 12832 family with sequence identity visualized using gradient colors. **d** Transmitted Y assemblies from father to sons in 14317 family. Pseudoautosomal regions (PARs), centromeres (Cen) and satellites (Sat), and X-transposed region (XTR) annotations were derived from Rhie et al. [71]. **e** Haplotype-resolved methylation at CpG islands (CpGIs) on the X chromosome in nine female-female quads. Mean methylation levels were calculated for each haplotype across 889 CpGIs and their ±5 kbp flanking regions on the X chromosome for 18 female individuals. Red denotes the maternal haplotype, while blue represents the paternal haplotype. A subset of individuals shows evidence of skewing of X inactivation. Source data are provided as a Source Data file.

(Coagulation Factor IX), or the intergenic region between *MXRA5* (Matrix Remodeling Associated 5) and *SNORA48B* (Small Nucleolar RNA, H/ACA Box 48B). These longer TR noncoding variants have only been observed in heterozygous states in females and variants of such lengths have yet to be observed in controls. Notably, the corresponding probands exhibited features commonly associated with

neurodevelopmental delay (Supplementary Fig. 9): IQ scores of 18, 13, and 20, and calibrated severity scores (CSS) of 9, 6, with the CSS data for the third not available.

Because LRS data allow CpG methylation to be robustly called[16], we used the 889 CpG islands across the X chromosomes to assess X chromosome inactivation (XCI) skewing in the blood of female

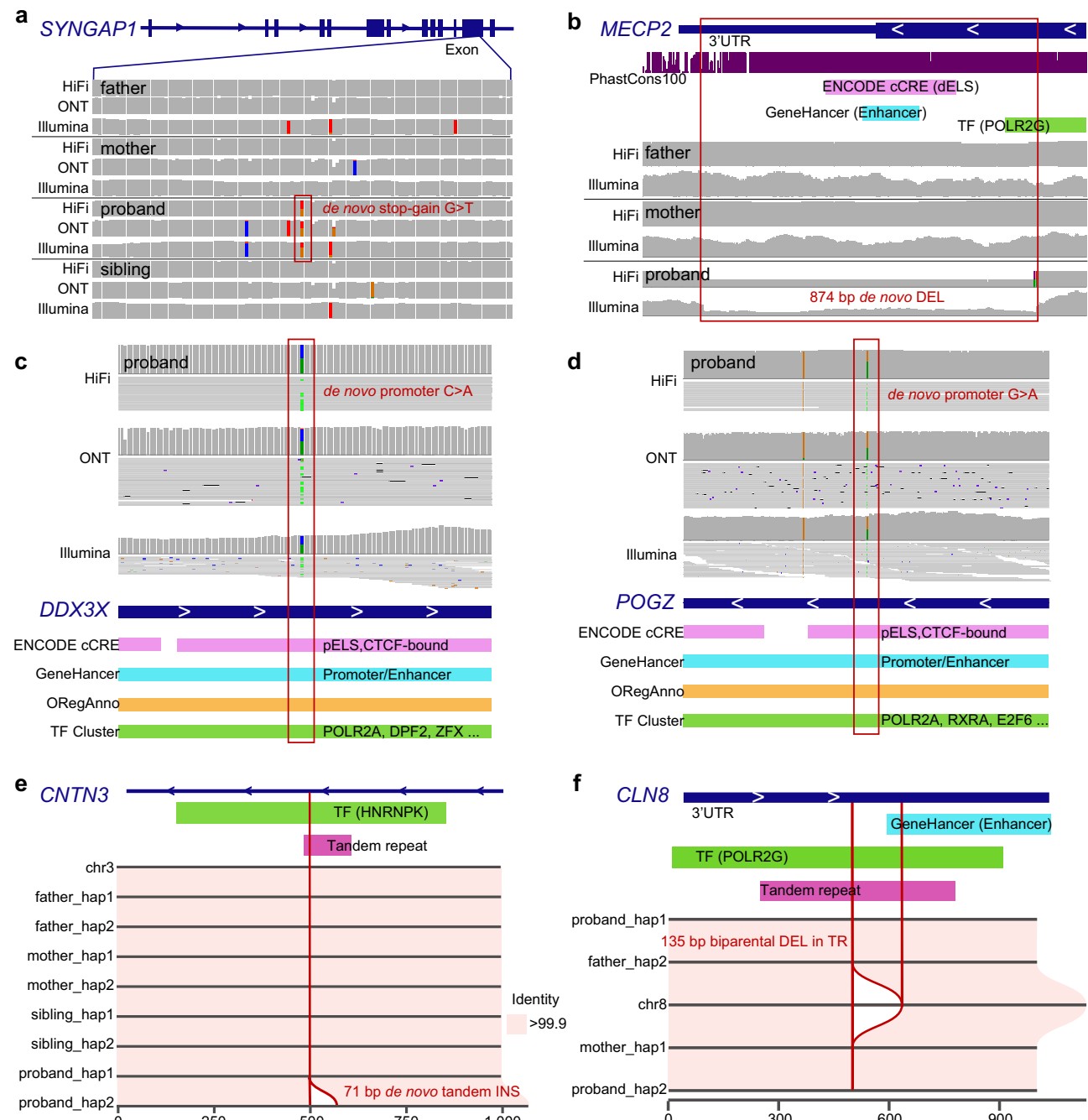

**Fig. 4 | Pathogenic and candidate variants missed by short-read WGS.** Long-read sequencing solved cases **a** (12237_p1) involving a stop-gain de novo mutation in *SYNGAP1* and **b** (HYZ207_p1) involving a de novo deletion in the last exon of *MECP2*. **c** A de novo candidate mutation in the promoter of *DDX3X* in 14133_p1. **d** A de novo candidate mutation in the promoter of *POGZ* in 12456_p1. **e** A 71 bp de novo tandem insertion in 11201_p1, predicted to interrupt the HNRNPK TF binding cluster in the intron of *CNTN3*. **f** A 135 bp homozygous tandem repeat (TR) contraction in the 3' UTR of *CLN8* in 11616_p1, predicted to disrupt the transcription of *CLN8*. The variants are highlighted with red boxes. ENCODE cCREs, ORegAnno, GeneHancer, ENCODE TF Clusters, and tandem repeats are published datasets from UCSC Genome Browser tracks. Source data are provided in Supplementary Data 4.

probands and their unaffected sisters. The analysis revealed extreme examples of XCI skewing, including preferential inactivation of the maternally inherited X chromosome (see 11071, Fig. 3e), potentially consistent with the mother carrying a damaged X chromosome.

**Pathogenic and autism candidate variant discovery**
Using the LRS and assembly data, we comprehensively searched for both pathogenic and potential candidate variants that were missed by SRS analyses (Methods). In total, we identified three DNMs classified as

pathogenic (Supplementary Data 4). Among the 46 idiopathic autism samples, we discovered a de novo stop-gain variant (G1124*) in *SYNGAP1* (Fig. 4a), a well-known autism-associated gene encoding a Ras GTPase-activating protein essential for synaptic function and cognitive development[46]. This pathogenic DNM in 12237_p1 is clearly supported by both HiFi and ONT reads from the proband but was not reported in three prior SRS analyses of this family[2,6,7]. An analysis of the Illumina sequence data, however, confirms the presence (Fig. 4a) of the variant in a GC-rich region of the genome where a cluster of rare and

**Table 1 | Summary of autism pathogenic and candidate variants**

| No. | Family ID | Sex | Gene(s) affected | Region | Variant type | Variant class | SRS status (prior study, caller or read support) |
|---|---|---|---|---|---|---|---|
| 1 | 12237 | F | *SYNGAP1*** | CDS | De novo stopgain | Pathogenic | Missing, yes, yes |
| 2 | HZRNM001 | F | *TBL1XR1*** | CDS | De novo missense | Pathogenic | Yes, yes, yes |
| 3 | HYZ207 | F | *MECP2*** | CDS | De novo 874 bp DEL | Pathogenic | Missing, yes, yes |
| 4 | 14133 | F | *DDX3X*** | Promoter | De novo substitution | Likely pathogenic | Yes, yes, yes |
| 5 | 12456 | F | *POGZ*** | Promoter | De novo substitution | Likely pathogenic | Yes, yes, yes |
| 6 | 11201 | M | *CNTN3** | TF binding | De novo 71 bp INS | Likely pathogenic | Missing, no, yes |
| 7 | 12651 | M | *LRPAP1* | 3'UTR | Biallelic 110 bp DEL | Likely pathogenic | Missing, no, no |
| 8 | 11611 | F | *CLN8** | 3'UTR | Biallelic 135 bp DEL | Likely pathogenic | Missing, no, no |
| 9 | 11918 | F | *TBC1D5** | TF binding | Biallelic 332 bp INS | Likely pathogenic | Missing, no, no |
| 10 | 12826 | F | *PREX1** | Enhancer | Biallelic 56 bp DEL | Likely pathogenic | Missing, no, no |
| 11 | 13414 | M | *ARHGEF10** | Promoter/Enhancer | Biallelic 193 bp DEL | Likely pathogenic | Missing, no, yes |
| 12 | 14350 | F | *LMF2, NCAPH2** | Promoter | Biallelic 90 bp DEL | Likely pathogenic | Missing, no, no |
| 13 | 14455 | F | *CPT1C* | Enhancer | De novo 73 bp INS | Uncertain | Missing, no, no |

*NDD candidate genes. ** High-confidence NDD or SFARI score 1 genes.

additional false calls were present, likely resulting in this region being subsequently filtered during QC.

Two additional pathogenic variants were discovered by LRS among the five autism-diagnosed females with features reminiscent of Rett syndrome. This included an 874 bp de novo DEL in *MECP2* in HYZ207_p1, which effectively disrupts the last exon of the gene and introduces a premature stop codon, truncating the protein by approximately 140 amino acids (Fig. 4b). Similar deletions involving exon 4 have been reported in Rett patients[47,48]. This pathogenic variant was previously missed in three rounds of clinical testing, including two gene panel sequencing tests through ARUP Laboratories and Quest Diagnostics, and one test of WES through Ambry Genetics. It was confidently identified in our LRS analysis and subsequently validated using all three sequencing platforms. We also identified a de novo missense mutation within *TBL1XR1* classified by ClinVar as pathogenic (rs1057517933). *TBL1XR1* encodes a transducin (beta)-like 1 X-linked receptor 1–that directly interacts with MECP2. This de novo *TBL1XR1* (D370N) missense variant was previously reported in the same patient[49] and two other cases in DECIPHER and was recently classified as a pathogenic variant[50]. We also observed a corresponding decrease in methylation at this CpG site within the exon of *TBL1XR1* (Supplementary Fig. 10). As part of this analysis, we also note two DNMs (Table 1) called by both SRS and LRS mapping to promoters of genes strongly implicated in neuronal development (*POGZ* and *DDX3X*; Fig. 4c, d). *POGZ* encodes a zinc finger protein involved in chromatin remodeling and transcriptional regulation[51], while *DDX3X*, an ATP-dependent RNA helicase, plays a crucial role in RNA metabolism, translation regulation, and neuronal development[52]. Both genes have been implicated in NDDs, including autism. These two DNMs have never been observed in gnomAD and given their critical location are candidates for functional testing using massively parallel reporter assay (MPRA) to determine if they significantly reduce expression levels.

Among de novo SVs, we identified several candidates of potential regulatory consequence (Table 1). For example, we identified a 71 bp de novo TR INS within the intron of *CNTN3*, a SFARI gene encoding Contactin 3, which mediates cell surface interactions during nervous system development and the outgrowth and guidance of axons and dendrites (Fig. 4e). This INS is predicted to disrupt HNRNPK transcription factor (TF) binding sites in 11201_p1; notably, HNRNPK also functions as an RNA-binding protein. We attempted to recall this SV using SRS-based callers, including Manta[53], Smoove (v0.2.5, https://github.com/brentp/smoove), CNVnator[54], and Canvas[55], and re-genotyped it using Paragraph[56] based on Illumina alignments. All SRS tools failed to detect this variant, likely due to its mapping within a CT-rich TR. We also identified a 73 bp de novo TR INS predicted to interrupt regulatory regions of *CPT1C* and TF binding clusters in 14455_p1. *CPT1C* encodes carnitine palmitoyltransferase 1C, a neuron-specific protein located in the endoplasmic reticulum, and has been associated with spastic paraplegia[57] with an emerging role in neuropsychiatric conditions[58]. This de novo INS was missed by all SRS-based callers, likely due to high-GC content (63% within 100 bp).

Finally, we evaluated rare biallelic inherited homozygous SVs for potential pathogenicity because of the unique capability of LRS to phase almost all variants[33]. We identified six candidates SVs inherited biparentally and associated with SFARI risk genes or cortex-specific regulatory regions (Table 1, Supplementary Data 4). None of these have been reported as homozygous in SRS genomic controls from gnomAD[31] or were identified in pangenome controls. This set includes: a homozygous DEL overlapping an enhancer located in the 3' untranslated region (UTR) of *CLN8* (a SFARI score 2 gene) in proband 11611_p1 (Fig. 4f); a homozygous DEL disrupting the enhancer of *ARHGEF10* (a SFARI score 2 gene), encoding a Rho guanine nucleotide exchange factor (GEF), in 13414_p1; a homozygous cortex-specific cis-regulatory element DEL in individual 12651_p1 mapping to the 3'UTR of *LRPAP1*, which encodes LDL receptor-related protein associated protein 1 and has been linked to dementia and late-onset Alzheimer's disease[59]; a homozygous DEL within an intronic enhancer of *PREX1* (a SFARI score 2 gene), in proband 12861_p1; a 332 bp homozygous INS disrupt TF binding in *TBC1D5* (a SFARI score 2 gene) in 11918_p1, and a 500 bp homozygous DEL mapping to the promoter region of both *LMF2* and *NCAPH2* (a SFARI score 3 gene) in individual 14350_p1 (Table 1). This promoter region is predicted to harbor regulatory activity based on multiple datasets, including chromatin accessibility and enhancer marks in the developing brain, suggesting potential cis-regulatory effects on the expression of one or both genes.

To assess the potential impact of candidate variants, we performed differential expression analysis using long-read RNA sequencing of lymphoblastoid cell lines generated with the PacBio Kinnex platform. Gene-level counts were obtained with IsoQuant[60] and analyzed with DESeq2[61] for probands carrying genes with two promoter mutations and five homozygous deletions that were expressed in blood. Across all seven candidate loci, no significant reduction in gene expression was detected in blood-derived RNA samples from the target proband compared to the remaining samples (Supplementary Fig. 11). The absence of detectable expression changes may reflect tissue-specific regulatory effects, as these variants are more likely to exert functional consequences in brain tissues, which are directly relevant to the disorder etiology.

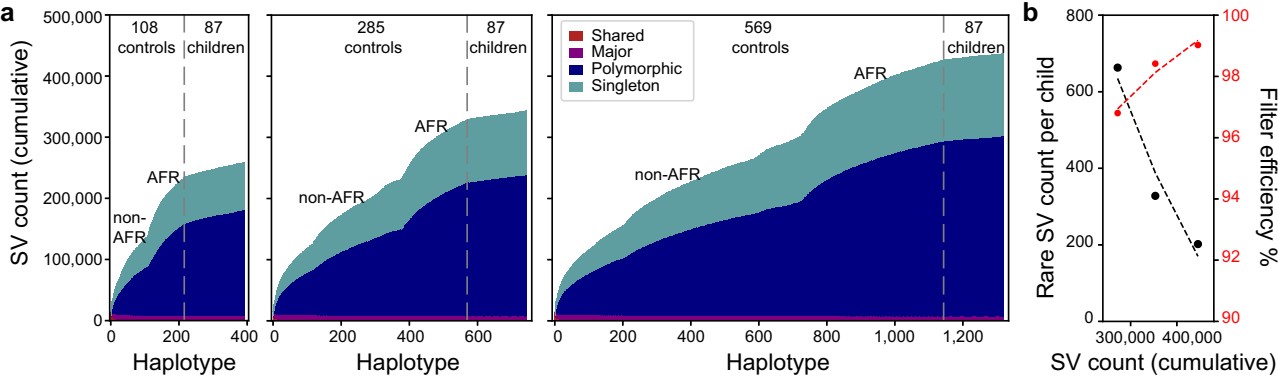

**Fig. 5 | Reduction of the rare SV pool with increasing control samples.**
**a** Cumulative discovery curves of SVs identified in different control cohorts of 108, 285 and 569 individuals, compared to 87 children (both affected and unaffected) from autism families (corresponding to Supplementary Data 2, 5 and 7, respectively). Control samples and discovery curves were computed for both African (AFR) and non-African (non-AFR) controls. **b** The inclusion of additional population controls refined the rare SV candidate pool, reducing the number of rare SVs (black) from 663 to 202, thus reducing the number of SVs under consideration from 97% to 99% (red). Source data are provided as a Source Data file.

## Pangenome increased sensitivity for pathogenic SV discovery

To enhance the power of pangenome filtering to effectively exclude common SVs and focus on a high-confidence pool of rare variants, we expanded the control cohort size from 108 to 285 and then to 569 individuals. The 285-control set contains 177 newly LRS and assembled samples from the HPRC (HPRCY2), while the 569-control set includes an additional 284 publicly available 1KGP samples sequenced using ONT (Supplementary Data 1, Supplementary Figs. 2 and 4). Increasing the number of controls, especially samples of African origin, nearly doubles the number of SVs from 271,375 (108 controls) to 445,142 (569 controls) nonredundant SVs (Supplementary Data 5–8). We generated a population-level SV reference and compared the MAF distribution between the 108 controls and the largest dataset comprising 569 controls and 102 unrelated parents ($n = 1342$). Based on this comparison, we estimated the fraction of variants captured at different MAF thresholds (Supplementary Fig. 12). Notably, only 16.2% of SVs with MAF < 0.1% in the larger dataset were detected in the 108-control set, illustrating the limited sensitivity of the smaller reference for identifying ultra-rare variants. In an ancestry- and sex-matched analysis restricted to individuals of European ancestry (38 autism families and 73 controls), each child carried an average of 754 rare SVs. Focusing on rare variants absent from the largest 569-control set, the number of rare SVs concomitantly drops to 202 events per sample (Fig. 5b), corresponding to ~74% of rare variants being filtered out by the more diverse population controls. This demonstrates the substantial gain in sensitivity for removing common and low-frequency variants when using a larger, heterogeneous reference panel. As a result, 99% of common SVs are excluded per individual forming a more tractable and potentially biomedically relevant set of rare variants for downstream interpretation and enrichment analyses. Although still not statistically significant ($p = 0.23$, two-sided Mann-Whitney U test), we note that applying 569 controls results in a larger difference with respect to SV burden between probands and unaffected siblings (Z = 1.20). Of note, only one of our previously 13 proposed candidates (73 bp de novo TR INS in *CPT1C*) was excluded and none of our pathogenic variants were excluded (Table 1, Supplementary Fig. 13).

We also compared our LRS SV datasets with a comprehensive SRS SV callset (INS and DEL) generated by GATK-SV[31] from 63,046 unrelated genomes, as represented in gnomADv4.1[31]. To ensure a fair comparison between SRS and LRS datasets that differ in resolution and breakpoint precision, we used relaxed matching criteria in Truvari bench: a minimum of 50% reciprocal overlap, 50% size similarity, a maximum breakpoint distance of 500 bp, and no sequence similarity requirement. We added the gnomAD SV IDs to the supplementary datasets (Supplementary Data 2, 5 and 7, Supplementary Fig. 14). Within the 108-control set, 162,916 of 271,375 total nonredundant SVs were observed in the autism families. Of these, 77.6% (126,377 SVs) were present in the 108 LRS controls, yielding 663 rare SVs per child. In contrast, only 24.1% (39,234 SVs) overlapped with the SRS-based gnomAD SVs from 63,046 unrelated controls (Supplementary Fig. 14), corresponding to 14,110 rare SVs per child when considering only short-read controls. Thus, despite its much larger sample size, the short-read dataset can be considered an auxiliary resource for further filtering rare SVs beyond those already excluded by the LRS controls, underscoring the substantially greater power of LRS data for rare variant filtering. Specifically, incorporating the SRS dataset would have further excluded 0.71% (147/20,716 per child) and 0.27% (55/20,716 per child) of SVs from the 108- and 569-control sets, respectively. Given the inherent differences in breakpoint resolution and variant representation between LRS and SRS platforms, these overlaps reflect potentially inflated estimates. A reassuring sign of the stringency and completeness of our filtering strategy is that all 12 candidates identified were rare variants absent from both the LRS and SRS control datasets.

## Discussion

SRS and microarray studies of autism families have estimated that as much as 30% of autism cases harbor a rare variant of large effect. While only approximately half of this burden has been discovered by SRS, it has been hypothesized that missing variants, as well as a portion of the missing heritability, may be attributed to impactful rare variants mapping in complex regions of the genome that are simply inaccessible or difficult to interpret using SRS approaches[42,62]. Targeted LRS studies for missing variants associated with Mendelian disease as well as select families with typically severe NDDs have suggested increases in diagnostic yield ranging from 7.3% to 33%[22–27,63]. Our analysis of 51 families mostly with daughters affected with autism (often more severely) where we attempted to sequence and assemble the entire euchromatic portion of each genome suggests a more modest rate of pathogenic variant discovery (5.9%). We consider this yield of recent pathogenic variant discovery low given that LRS application increased genome-wide sensitivity of DNM detection by 20–40%[39] and we purposefully sequence affected females where the probability of discovery of a large effect mutation is expected to be higher[64]. It should be noted that LRS alignment to a reference was sufficient to detect these variants previously missed by SRS. Genome assemblies, while helpful for parent-of-origin determination and phasing variants, were not critical to the discovery of most variants.

In the end, we identified three pathogenic variants, including only one daughter previously classified as idiopathic (*SYNGAP1*), while the other two pathogenic mutations (a de novo disruptive missense mutation in *TBL1XR1* and de novo SV affecting the last exon of *MECP2*) arose in daughters suspected of Rett syndrome. A retrospective analysis of whole-genome SRS data confirmed the presence of the variants although in two of the three cases the variants would have been challenging to call without LRS. In addition, we identified nine additional candidate mutations (17.6% of patients) for further functional testing. The majority of these (7/9) were SVs that would have been missed by most standard SRS-based SV callers (Table 1). In contrast to DNMs, these SV mutations did not map to coding regions but instead were inherited and corresponded to homozygous deletions or insertions within regulatory DNA often for genes associated with autism or neurodevelopment. The LRS data provided unambiguous phasing allowing rare biallelic inherited homozygous events to be discovered and characterized. These findings may suggest that some fraction of autism arises as a result of recessive or a contribution of oligogenic mutations[2,65]. Advanced sequencing techniques such as LRS will be required to reveal the full spectrum of mutations contributing to autism.

Compared to earlier work with large CNVs[9], we do not yet observe a significant increase in SV burden when comparing affected and unaffected siblings (Fig. 2). This lack of statistical significance is likely due to the limited sample size of this study. Notwithstanding, there are some interesting trends. For example, we note a slight excess of deletions on the X chromosome among affected daughters when compared to their unaffected sisters. Indeed, our ability to sequence and assemble ~96% of the X chromosome, as well as most of the euchromatic portion of the Y chromosome (Fig. 3), will be critical for evaluating the contribution of sex chromosomes to autism sex bias. Sex chromosomes are routinely excluded from whole-genome SRS studies because of ploidy issues and challenges with repetitive regions[6,7,66]. LRS and assembly largely overcome these limitations. Moreover, the ability to concurrently assay methylation status of CpG islands genome-wide and readily distinguish X chromosome skewing patterns (Fig. 3) will also advance the discovery of epigenetic mutations as well as potentially damaged X chromosomes as more and more autism genomes are sequenced.

Perhaps, most importantly, has been the ability to leverage pangenomes[30] to restrict the focus of SV discovery to variants that are private to autism families. Unlike SNVs mapping to coding sequencing, databases such as gnomAD[31,32,67] are largely incomplete for variants, especially SVs, mapping to more complex regions of the genome. A typical human genome harbors over 25,000 variants while whole-genome SRS has been shown to reliably report only 11,000 such variants[15]. In this study, we used more completely sequenced and assembled genomes from public initiatives such as HGSVC, HPRC and 1KGP ONT[18,29,30,42] as controls to filter out more common SVs. Using 569 LRS control genomes identifies 445,142 nonredundant SVs in total. Under a model of ultra-rare SVs contributing to disease, we, as a result, essentially exclude 99% of the more common variants allowing us to focus on 202 private or de novo SVs per child. We note that the difference with respect to SV burden increases between the proband and the unaffected sibling, although it does not yet reach statistical significance. If we further restrict this analysis to SVs corresponding to regions of the genome under functional constraint (with Gnocchi ≥ 4[32] and overlap with predicted promoters), this set would further reduce to one or two SVs per genome. As the human pangenome continues to grow and more complete genetic information emerges, the potential to discover variants of pathogenic significance will increase.

## Methods

### Sample selection and previous sequence characterization

Illumina WGS was previously applied to the blood DNA of the 174 individuals corresponding to 46 families with idiopathic autism from the SSC and SAGE. Potential pathogenic variants were screened from SRS based on the studies by Wilfert et al. [2] and Fu et al. [6], including (1) SNVs: de novo likely gene-disruptive (LGD), de novo missense, and rare inherited LGD variants in NDD-related genes or genes with pLI scores ≥ 0.9; (2) CNVs in the morbidity map[4] or span of a gene with pLI score ≥ 0.9. No known genetic cause was identified in the 46 probands from SRS data[2,4,6,35]. In addition, the 46 probands selected in this study do not exhibit exceptional polygenic risk scores among the 13,989 individuals examined[2] and have known IQs ranging from 13 to 91. Similarly, five girls diagnosed with Rett-like syndrome from Baylor College of Medicine had no causal variants identified in *MECP2* by prior gene panel testing or WES (Supplementary Data 1). The selected families were predominantly those with female probands due to the interest in discovering X chromosome variants, and the large-effect variants are more likely to be discovered. As shown in Fig. 1a, 17 quads with sex-matched offspring (12 female-female and 5 male-male quads), 19 quads with sex-mismatched offspring (16 female-male and 3 male-female quads), and 17 trios (15 female and 2 male trios) were included in this study. Briefly, the 51 probands (41 females and 10 males) were selected to represent cases that pose difficulties in pinpointing the cause of autism using Illumina SRS data.

### LRS data generation and QC

We generated PacBio HiFi and ONT sequencing data at the University of Washington (UW) for 189 individuals. Illumina WGS for Rett-like trios was generated from blood DNA using the TruSeq library kit and sequenced on a NovaSeq with paired-end 150 bp reads at the Northwest Genomics Center. For the five Rett-like trios, DNA was extracted from blood using the Monarch HMW DNA Extraction Kit for Cells & Blood from NEB (T3050L) (*n* = 6) or the Qiagen Puregene Blood Core Kit (158023) (*n* = 9), following the manufacturer's specifications. The whole-blood DNA from the SSC was extracted previously as part of that biobank. The cell line DNA was extracted from lymphoblastoid cell lines with either a modified Gentra Puregene (Qiagen) protocol when used for ONT sequencing or with the Monarch HMW kit (NEB T3050L) when used for PacBio sequencing. ONT libraries were constructed using the Ligation Sequencing Kit (ONT, LSK110 and LSK114) with modifications to the manufacturer's protocol. The library was loaded onto a primed R9.4.1 or R10.4.1 flow cell (FLO-PRO002 or FLO-PRO114M) for sequencing on the PromethION, with two nuclease washes and reloads after 24 and 48 hours of sequencing.

PacBio HiFi data from family 14455 (*n* = 4) were published in Noyes et al. [17]. Data from three families (*n* = 11) were graciously generated by PacBio. Remaining individuals' HiFi data were generated from blood or cell line HMW DNA according to the manufacturer's recommendations. At all steps, quantification was performed with Qubit dsDNA HS (Thermo Fisher Scientific, Q32854) measured on DS-11 FX (Denovix) with the size distribution checked using FEMTO Pulse (Agilent, M5330AA and FP-1002-0275.) The samples' incoming size distribution determined shearing conditions, either no shear (*n* = 12), or sheared with the Megaruptor 3 (Hologic Diagenode, B06010003 & E07010003) system using one (*n* = 36) or two (*n* = 130) sequential runs to target a peak size of ~20 kbp. After shearing, the DNA were used to generate PacBio HiFi libraries using the Express Template Prep Kit v2 (*n* = 12, PacBio, 100-938-900) or SMRTbell prep kit 3.0 (*n* = 166, PacBio, 102-182-700). Size selection was performed with Pippin HT using a high-pass cut-off between 9-17 kbp based on shear size (Sage Science, HTP0001 and HPE7510). Libraries were sequenced either on the Sequel II platform on SMRT Cells 8 M (PacBio, 101-389-001) using Sequel II sequencing chemistry 2.0 (*n* = 16, PacBio, 101-842-900), 2.2 (*n* = 4, PacBio, 102-089-000), or 3.2 (*n* = 32, PacBio,102-333-300) with 2 h pre-extension and 30 h movies on SMRT Link v9-11.1, or on the Revio platform on Revio SMRT Cells (PacBio, 102-202-200) and Revio polymerase kit v1 (*n* = 126, PacBio, 102-817-600) with 2 h pre-extension and 24 or 30 h movies on SMRT Link v12.0-13.1.

To ensure the use of high-quality reads for constructing robust assemblies, we first filtered out nonhuman contamination reads from both HiFi and ONT data. We employed highly accurate Illumina reads and utilized yak (commit f389bad, https://github.com/lh3/yak.git) to calculate the QV for each read, and a two-sided z-test was conducted on the resulting QV values. Reads with a z-score less than −2, indicating a potential risk of contamination, were compared against the Kraken2[23] (v2.1.3) database, and those identified as nonhuman in origin were excluded. We masked the Y chromosome from GRCh38 to generate the GRCh38noY reference genome. HiFi reads were aligned to GRCh38 for males and to GRCh38noY for females using pbmm2 (v1.13.1, https://github.com/PacificBiosciences/pbmm2). To ensure the family pedigree, the relatedness between sample pairs and ancestry prediction were conducted using Somalier[68] (v0.2.19) based on the alignment (Supplementary Data 1). ONT reads were aligned to the reference genome using minimap2 (v2.28.0), and the family pedigree was confirmed with VerifyBamID (v2.0.1). In addition, ntsm (v1.2.1) was applied to each HiFi and ONT fastq for sample swap detection.

## Phased genome assembly construction

The HiFi assembly was constructed by hifiasm[36] (v0.16.1). Parental short reads were processed with yak (v0.1, https://github.com/lh3/yak.git) and then hifiasm trio-binning mode was used for phasing child samples, while the parental assemblies were partially phased by default. The sex chromosome contigs from father samples were aligned to the T2T-CHM13v2.0 reference to reassign the Y chromosome contigs to hap1 (or paternal haplotype) and X chromosome contigs to hap2 (or maternal haplotype). Assembly QVs were evaluated by merqury (v1.3) with k-mers from Illumina data (meryl v1.4); next, the completeness of phased assemblies relative to the reference and contig N50 values were calculated (Supplementary Data 1).

## Variant discovery

SNVs and small indels in 73 children, with available ONT data, were recalled with DeepVariant[38] (v1.4.0) and GATK[37] (v4.3.0.0) based on the HiFi alignments to the T2T-CHM13v2.0 reference. DNMs including both de novo and postzygotic mutations were further validated by ONT and/or Illumina reads using the method described in Noyes et al. [39]. We annotated DNMs using the Ensembl Variant Effect Predictor (VEP, v110.1) and referred to the predicted impact and scores from dbNSFP (v4.8a), CADD score (v1.3), and gnomAD genome allele frequency[32] (v4.1.0) by lifting coordinates over to both GRCh38 and GRCh37 using UCSC LiftOver (Supplementary Data 4). SVs were called using the phased assembly variant caller (PAV[18], v2.3.4) by aligning the assembled genomes to a reference (GRCh38 or GRCh38noY). The alignment-based SVs were called by PacBio SV calling and analysis tool (PBSV, v2.9.0, https://github.com/PacificBiosciences/pbsv) and Sniffles[40] (v2.2). SVs from the control population were additionally called by Delly (v1.2.6), Sawfish (v0.12.4), and cuteSV (v2.1.0). The ratio of insertions to deletions, the ratio of heterozygous to homozygous variants, and the size distribution of SVs were evaluated for each VCF file.

## SV merging and filtering strategy

The strategy involved three major steps. Briefly,

1. Callerset validation. SVs detected in each sample from different SV callers were first normalized, sorted, and merged with BCFtools (v1.20) on the basis of PAV callset, and then collapsed by Truvari (v4.3.1). SVs located within known genomic gaps, telomeric regions, centromeres, and PARs on GRCh38 from UCSC Genome Browser tracks were excluded. We used the following command lines:

bcftools merge--thread {threads} --merge none --force-samples -O z -o {output.vcf.gz} {input.vcf1.gz} {input.vcf2.gz} {input.vcf3.gz}

truvari collapse -i {input.vcf.gz} -c {output.removed.vcf.gz} --sizemin 0 --sizemax 1000000 -k maxqual --gt het --intra --pctseq 0.90 --pctsize 0.90 --refdist 500 | bcftools sort --max-mem 8 G -O z -o {output.collapsed.vcf.gz}

2. Inter-sample merge. We extracted SVs supported by PAV and at least one of the alignment-based callers for each individual and then merged SVs from both controls and autism families using a list of VCFs with Truvari. In this study, we have applied three control sets consisting of 108, 285, and 569 individuals using the following command lines:

bcftools merge --threads {threads} --merge none --force-samples --file-list {input.vcflist} -O z | bcftools norm --threads {threads} --do-not-normalize --multiallelics --any --output-type z -o {output.mergevcf.gz}

truvari collapse --input {input.mergevcf.gz} --collapsed-output {output.removed_vcf.gz} --sizemin 0 --sizemax 1000000 --pctseq 0.90 --pctsize 0.90 --keep common --gt all | bcftools sort --max-mem {resources}G --output-type z > {output.collapsed_vcf.gz}

3. Rare SV pool discovery. We developed a custom script to extract six categories of rare SVs as described in the main text (v1.0.0, https://github.com/EichlerLab/asap). For autosomal SVs, we retained heterozygous and homozygous SVs present only in the children but not in the controls. For SVs on the sex chromosomes, we performed sex-matched comparisons and filtered SVs seen in controls with the same sex.

Separately, we compared our LRS SV datasets with the SRS control set from 63,046 unrelated genomes in gnomAD v4.1 to evaluate the overlap between platforms. SV comparison was conducted using Truvari bench with relaxed matching parameters to account for differences in breakpoint precision between LRS and SRS datasets:

truvari bench -c {gnomad.v4.1.INSDEL50.non_neuro_controls.sites.vcf.gz} -b {108/285/569ctr_189asd_collapsed.vcf.gz} --pctsize 0.5 --pctseq 0.0 --pctovl 0.5 --sizefilt 50 --sizemax 100000 -o {output}

We incorporated gnomAD SV IDs, overlap metrics (PctSizeSimilarity:PctRecOverlap:SizeDiff), and allele counts (heterozygous, homozygous, and sex-specific; overall and within the non-neuro subset into the collapsed SV tables (Supplementary Data 2, 5 and 7)). Similar to step 3 above, we applied sex- and zygosity-aware filtering to identify rare SVs absent from all SRS genomes.

## Transmission curation of rare SVs

Parental genotypes corresponding to each rare SV in the children are provided in Supplementary Data 2, 3, and 5–8. Variants that followed Mendelian inheritance patterns were designated as high-confidence SVs. The remaining SVs were subjected to transmission curation using the following toolchain:

1. Initial caller support using Truvari. To minimize the loss of inheritance information in parents lacking SV caller support, we collapsed SVs from parents with the child's SVs using truvari bench,

truvari bench -c {fa,mo}.vcf.gz -b child.vcf --pctsize 0.9 --pctseq 0.9 -o {fa_child,mo_child}

2. Callable region evaluation using BoostSV (v1.0). To ensure the SVs fall within confidently callable regions across samples in a single family, we developed a tool, BoostSV (v1.0, https://github.com/jiadong324/BoostSV), leveraging a machine-learning approach trained on control samples[33]. This tool evaluates read support, mapping quality, and data quality metrics from alignments surrounding the target SVs in each parent. A quality threshold of 0.5 was applied to obtain the transmission.

3. Genotyping support using kanpig. We applied the k-mer-based genotyper kanpig (v0.3.1) to parental HiFi alignments for each SV to assess allele presence and genotype consistency.

4. Rare TR expansions/contractions and multiple sequence alignment (MSA). SVs overlapped with TR catalogs derived from the four-generation control family in Porubsky et al. [33] were genotyped by TRGT[69] (v1.4.1) using HiFi alignments with the following command:

trgt genotype --genome {input.ref} --repeats {input.bed} --reads {input.bam} -t {threads} --output-prefix {wildcards.sample}/trgt --karyotype {XX/XY}

Genotype (GT) and allele length (AL) information were extracted from the TRGT output across all individuals of the autism family to detect transmission or TR outliers. For complex TR motif structures, we validated the SVs in the target sequence from assemblies using the MSA approach,

# Align assemblies to the reference using minimap2 (v2.28.0):

minimap2 -c -t {threads} -K {resources.mem}G --cs -x asm20 -m 10000 -z 10000,50 -r 50000 --end-bonus=100 -O 5,56 -E 4,1 -B 5 --secondary=no -eqx -Y {input.ref} {input.asm} > {output.paf}

# Liftover target sequence coordinates onto query sequence using rustybam (v0.1.33, https://github.com/mrvollger/rustybam) and extract the target sequence using SAMtools (v1.16.1):

rustybam liftover --bed {input.bed} {input.paf} > {output.liftover.paf}

samtools faidx {input.asm} {liftover.paf.query_region} > {output.fa}
#. MSA were performed using MAFFT (v7.525) and visualized in Jalview (v2.11.4.1):

mafft --adjustdirection --thread {threads} --auto --reorder {input.combined.fa} > {output.msa.fa}

5. Read-based support validation using subseq. To further assess SV transmission, subseq[18] (v1.0, https://github.com/EichlerLab/subseq-smk) was used to quantify read support for each SV in parental genomes. A dynamic window size was determined based on the SV size, and the number of reads traversing the window were counted. The {size50_1_1} parameter was used, requiring a minimum of one read supporting the SV while allowing a 50% size deviation penalty.

6. Manual inspection using IGV. For SVs lacking sufficient support in Steps 1-5, we conducted visual inspection of supporting reads in the Integrated Genomics Viewer (IGV, v2.16.0). Evidence from HiFi alignments, HiFi assemblies, Illumina alignments, and, when available, ONT alignments (186 samples) was reviewed to further assess inheritance status and assign low-confidence transmissions.

## SV annotation

A customized script was implemented to annotate previously published NDD candidate genes and regulatory elements, as well as integrate annotations from AnnotSV (v3.4). In terms of REG, we integrated published datasets from UCSC Genome Browser tracks, including candidate cis-regulatory elements (ENCODE Regulation, ENCODE cCREs, ORegAnno, GeneHancer) and ENCODE histone marks (H3K27Ac, H3K4Me1, H3K4Me3). Additionally, we incorporated epigenomic profiles from the cerebral cortex (brainREG), particularly a cis-regulatory element map generated from 27 male and 21 female prenatal human cortex samples by ATAC-seq and consensus maps for CTCF, H3K27ac, H3K27me3, and H3K4me3 generated from six male and five female prenatal human cortex samples by CUT&Tag[44]. For NDD candidate genes, we primarily focused on those previously reported[5–7,45]. Additional ENCODE TF Clusters, UCSC noncoding RNA (tRNA, snRNA, lincRNA, sno_miRNA), repetitive regions (UCSC SegDup, UCSC RepeatMasker, UCSC Simple Repeats, TRs[33]), noncoding constraint Gnocchi score[32] and CADD-SV score (v1.1.2) were annotated.

Potential pathogenic variants were confirmed with gnomAD allele frequency and the 569-control dataset. SVs and CNVs from Illumina WGS data for the selected samples were recalled by Manta[53] (v.1.5.0), Smoove (v0.2.5, https://github.com/brentp/smoove), CNVnator[54] (v0.3.3), Canvas[55] (v1.40.0.1613+master) and genotyped using Paragraph[56] (v2.4) to evaluate SRS detection.

## Sex chromosome assemblies and transmission

We partitioned the sex chromosomes (T2T-CHM13v2.0, excluding the PAR, centromere, and highly repetitive Yq12 heterochromatin regions) into 1 Mbp windows and identified those covered by contigs that aligned to ≥95% of the window sequence with no more than three overlapping contigs. The coverage percentage was calculated as the number of qualified windows divided by the total number of windows,

representing coverage relative to the reference sex chromosomes (Fig. 3a, b, Supplementary Fig. 8). We designed a pipeline to assemble contiguous X and Y chromosomes and validate transmission patterns within families (https://github.com/projectoriented/contiguous-X). The pipeline consists of two main steps: (1) scaffolding contigs with at least 50% of their sequence aligning to chromosome X (minimap2) via RagTag (v2.1.0), and (2) visualizing alignments across haplotypes within the family using the SVbyEye R package.

## Methylation analysis

We developed a pipeline to extract phased methylation signals from ONT alignments (https://github.com/projectoriented/continuous-methylation). Briefly, when parental Illumina reads were available, ONT reads from the offspring were phased using the Canu (v2.1.1) trio-binning method. The phased ONT reads were then aligned to the GRCh38 reference genome using minimap2 (v2.24.0) and haplotagged accordingly. For individuals (e.g., parents) without available parental Illumina data, SVs were identified using Sniffles[40] (v2.2) and small variants were called with Clair3 (v1.0.2), followed by haplotagging with LongPhase (v1.7.2). Methylation tags from unmapped BAM files were linked to the phased alignments using methylink (v0.6.0, https://github.com/projectoriented/methylink). Finally, Modkit (v0.3.1, https://github.com/nanoporetech/modkit) was used to generate methylation calls in BED format via pileup function. We analyzed mean methylation differences between the two X chromosome haplotypes across 889 CpG islands, including ±5 kbp flanking regions. We then used deepTools (v3.5.5, https://github.com/deeptools/deepTools) to compute the methylation matrix and generate the plots using the following parameters:

computeMatrix scale-regions -R {hg38XcpgI.bed} -S {hap1.bigWig} {hap2.bigWig} -b 5000 --regionBodyLength 2000 -a 5000 --skipZeros --numberOfProcessors 12 --outFileName {matrix}; plotProfile -m {matrix} --averageType mean --samplesLabel pat mat --startLabel CpGI --endLabel CpGI --yAxisLabel "Mean of methylation%" --yMin 0 --yMax 100 --colors blue red --legendLocation lower-right --perGroup --plotType=lines --plotTitle {sample} --dpi 600 --outFileName {plot}

## Long-read Kinnex generation and analysis

Total RNA was extracted from 5 million lymphoblast cells using the RNeasy mini kit from Qiagen. After initial QC using DS-11 FX (Denovix) for UV-Vis quantification and Bioanalyzer RNA 6000 Nano (Agilent, P/N G2939A and 5067-1511) for quality score calculation, 150–250 ng per sample were processed into cDNA and sequencing library concatamers using the Iso-Seq Express 2.0 and Kinnex full-length RNA kits (PacBio, P/N 103-071-500 and 103-072-000) according to manufacturer's protocols with the following modifications: amplified cDNAs were analyzed for average size using Femto Pulse (Agilent P/N M5330AA and FP-1003-0275) and quantified using fluorescence (Qubit HS DNA Fisher Scientific P/N Q32854) before equimolar pooling and an additional 1.1X SMRTbell Cleanup Bead wash to remove residual primer-dimer before proceeding to Kinnex PCR. Kinnex concatamers were checked for length using Femto Pulse (P/N FP-1002-0275) before loading on 1 SMRT Cell 25 M on the PacBio Revio sequencing platform using SPRQ chemistry with Adaptive Loading and a 30-hour acquisition time. After sequencing, raw data were processed with the Read Segmentation and Iso-Seq workflow in SMRT Link v25.3 using default parameters to generate sample-demultiplexed full-length reads (flnc.bam) files.

BAM files were converted to FASTA format using SAMtools (v1.21) and aligned to the GRCh38 reference genome with minimap2 (v2.28) using the parameters -ax splice:hq -uf --secondary=no --eqx -K 200 M. Gene and transcript quantifications were performed with IsoQuant[60] (v3.10.0) using GENCODE v49 annotations. Raw gene counts from the target proband and remaining samples were used as case and control groups, respectively, for differential expression analysis with DESeq2[61] (v1.50.0), incorporating sex and diagnosis as variables.

## Ethics

Ethical approval for this study was granted by the University of Washington IRB Committee B, under STUDY ID: STUDY00000383.

## Reporting summary

Further information on research design is available in the Nature Portfolio Reporting Summary linked to this article.

## Data availability

The underlying sequencing data, as well as the processed assembly and alignment files used for analysis in this study for the SSC samples (n = 168) and the complete sample set (n = 189), are available to approved researchers through SFARI Base under Dataset ID DS0000104 and through the National Institute of Mental Health Data Archive (NDA) under Collection ID 3780. Source data are provided with this paper.

## Code availability

The code used to perform the analyses and generate results in this study is publicly available and has been deposited in GitHub at https://github.com/EichlerLab/asap, under the MIT license. The specific version of the code associated with this publication is archived in Zenodo and is accessible via https://doi.org/10.5281/zenodo.18149644[70]. A full list of all software used, along with references, is provided in the Supplementary Notes and Supplementary References. Any additional information required to reanalyze the data reported in this work is available from the lead contact upon request.

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

## Acknowledgements

We thank all of the individuals who participated in this research. We also thank all contributing investigators to the consortia datasets used here from SSC, SAGE, and Baylor College of Medicine, and the families who participated in this research, without whose contributions, genetic studies would be impossible. We thank Tonia Brown for assistance in editing this manuscript. We thank Tom Mokveld (PacBio) for helpful discussions. This work was supported, in part, by the US National Institutes of Health (NIH R01MH101221 to E.E.E.; R01NS057819 to H.Y.Z.; 1F32HD116501-01 to R.M-S.; and DP5OD033357 to D.E.M.) and the Simons Foundation (SFARI #810018 to E.E.E., H.Y.Z., T.J.N., and A.C.). E.E.E. and H.Y.Z. are investigators of the Howard Hughes Medical Institute. This work was also supported, in part, by the National Natural Science Foundation of China (82201314 and 82471194) to T.W. We would like to acknowledge the National Genome Research Institute (NHGRI) for funding the following grants supporting the creation of the human pangenome reference: U41HG010972, U01HG010971, U01HG013760, U01HG013755, U01HG013748, U01HG013744, R01HG011274, and the Human Pangenome Reference Consortium (BioProject ID: PRJNA730823). This article is subject to HHMI's Open Access to Publications policy. HHMI lab heads have previously granted a nonexclusive CC BY 4.0 license to the public and a sublicensable license to HHMI in their research articles. Pursuant to those licenses, the author-accepted manuscript of this article can be made freely available under a CC BY 4.0 license immediately upon publication.

## Author contributions

Y.S. and E.E.E. conceptualized the study. K. Ho., R.A.P.P., D.P., and B.S. collected samples. K.M.M., K. Ho., G.H.G., and J.K. generated the data. Y.S., Y.K., I.W., and N.K. performed data quality control. Y.S., J.L., M.D.N., I.W., and N.K. conducted the formal analyses. Y.S. and I.W. created the visualizations. Y.S., J.L., W.T.H., and M.W. developed the methodology. K. He., D.K., J.A.G., D.E.M., and the HPRC provided resources. Y.K., I.W., N.K., and J.W. developed software. Y.S., J.L., I.W., T.W., R.A.P.P., R.M.-S., and F.C. performed validation. Y.S. wrote the original draft. Y.S., E.E.E.,

H.Y.Z., J.L., M.D.N., Y.K., I.W., N.K., W.T.H., M.W., J.W., K. Ho., K.M.M., G.H.G., J.K., T.W., K. He., D.K., R.A.P.P., R.M.-S., F.C., D.P., B.S., J.A.G., D.E.M., T.J.N., A.C., H.B.-R. and the HPRC reviewed and edited the manuscript. E.E.E., H.Y.Z., A.C., and T.J.N. supervised the study.

## Competing interests

E.E.E. is a scientific advisory board (SAB) member of Variant Bio, Inc. D.E.M. is on SABs at Oxford Nanopore Technologies (ONT) and Basis Genetics, is engaged in research agreements with ONT and PacBio, has received research and travel support from ONT and PacBio, holds stock options in MyOme and Basis Genetics, and is a consultant for MyOme. J.A.G. has received travel support from ONT. H.Y.Z. is a member of the Regeneron Board of Directors and an advisory board member to The Column Group, Cajal Therapeutics (also co-founder), and Lyterian. D.P. provides consulting service to Ionis Pharmaceuticals, M2DS Therapeutics and Acadia Pharmaceuticals. All other authors declare no competing interests.

## Additional information

[1]Department of Genome Sciences, University of Washington School of Medicine, Seattle, WA, USA. [2]Department of Medical Genetics, Center for Medical Genetics, School of Basic Medical Sciences, Peking University Health Science Center, Beijing, China. [3]Neuroscience Research Institute, Peking University; Key Laboratory for Neuroscience, Ministry of Education of China & National Health Commission of China, Beijing, China. [4]Autism Research Center, Peking University Health Science Center, Beijing, China. [5]Department of Neurological Surgery, University of California, San Francisco, CA, USA. [6]Weill Institute for Neurosciences, University of California San Francisco, San Francisco, CA, USA. [7]Center for Human Genetics and Genomics, New York University Grossman School of Medicine, New York, NY, USA. [8]Department of Human and Molecular Genetics, Baylor College of Medicine, Houston, TX, USA. [9]Jan and Dan Duncan Neurological Research Institute at Texas Children's Hospital, Houston, TX, USA. [10]Texas Children's Hospital, Houston, TX, USA. [11]Section of Pediatric Neurology and Developmental Neuroscience, Department of Pediatrics, Baylor College of Medicine, Houston, TX, USA. [12]Division of Genetic Medicine, Department of Pediatrics, University of Washington, Seattle, WA, USA. [13]Molecular and Cellular Biology Program, University of Washington, Seattle, Washington, USA. [14]Department of Laboratory Medicine and Pathology, University of Washington, Seattle, WA, USA. [15]Department of Neuroscience and Physiology, New York University Grossman School of Medicine, New York, NY, USA. [16]Department of Pediatrics, Baylor College of Medicine, Houston, TX, USA. [17]Department of Neuroscience, Baylor College of Medicine, Houston, TX, USA. [18]Department of Neurology, Baylor College of Medicine, Houston, TX, USA. [19]Howard Hughes Medical Institute, Baylor College of Medicine, Houston, TX, USA. [20]Howard Hughes Medical Institute, University of Washington, Seattle, WA, USA. ✉e-mail: ee3@uw.edu

## Human Pangenome Reference Consortium (HPRC)

Derek Albracht[21], Ivan A. Alexandrov[22], Jamie Allen[23], Alawi A. Alsheikh-Ali[24], Nicolas Altemose[25], Casey Andrews[26], Dmitry Antipov[27], Lucinda Antonacci-Fulton[21], Mobin Asri[28], Marcelo Ayllon[1], Jennifer R. Balacco[29], Floris P. Barthel[30], Halle D. Bender[28], Andrew P. Blair[28], Davide Bolognini[31], Katherine E. Bonini[32], Christina Boucher[33], Guillaume Bourque[34,35,36], Silvia Buonaiuto[37], Shuo Cao[37], Andrew Carroll[38], Ann M. Mc Cartney[28], Monika Cechova[28], Pi-Chuan Chang[38], Xian Chang[28], Jitender Cheema[23], Haoyu Cheng[39], Claudio Ciofi[40], Hiram Clawson[28], Sarah Cody[21], Vincenza Colonna[37], Holland C. Conwell[41], Robert Cook-Deegan[42], Mark Diekhans[28], Maria Angela Diroma[40], Daniel Doerr[43,44,45], Zheng Dong[26], Danilo Dubocanin[25], Richard Durbin[46,47], Jana Ebler[43,48], Evan E. Eichler[1,20,49], Jordan M. Eizenga[28], Parsa Eskandar[28], Eddie Ferro[33], Anna-Sophie Fiston-Lavier[50,51], Sarah M. Ford[41], Willard W. Ford[52], Giulio Formenti[29], Adam Frankish[23], Mallory A. Freeberg[23], Qichen Fu[26], Stephanie M. Fullerton[53], Robert S. Fulton[21], Yan Gao[54], Gage H. Garcia[1], Obed A. Garcia[55], Joshua M. V. Gardner[28], Shilpa Garg[56], Erik Garrison[37], Nanibaa' A. Garrison[57,58,59], John E. Garza[21], Margarita Geleta[60], Mohammadmersad Ghorbani[61], Tina A. Graves-Lindsay[21], Richard E. Green[41], Cristian Groza[62], Andrea Guarracino[30,37], Melissa Gymrek[63], Maximilian Haeussler[28], Leanne Haggerty[23], Ira M. Hall[64,65], Nancy F. Hansen[27], Yue Hao[30], Mohammad Amiruddin Hashmi[24], David Haussler[28], Prajna Hebbar[28], Peter Heringer[43,44,45], Glenn Hickey[28], Todd L. Hillaker[28], S. Nakib Hossain[23], Neng Huang[54,66], Sarah E. Hunt[23], Toby Hunt[23], Alexander G. Ioannidis[25,28], Nafiseh Jafarzadeh[28], Nivesh Jain[29], Erich D. Jarvis[29,49],

Maryam Jehangir[30], Juan Jiang[26], Edward A. Belter Jr[21], Jonathan LoTempio Jr[67], Eimear E. Kenny[32], Juhyun Kim[27], Bonhwang Koo[29], Sergey Koren[27], Milinn Kremitzki[21,26], Charles H. Langley[68], Ben Langmead[69], Heather A. Lawson[26], Daofeng Li[26], Heng Li[54,66], Wen-Wei Liao[64,65], Jiadong Lin[1], Tianjie Liu[26], Glennis A. Logsdon[70], Ryan Lorig-Roach[28], Hailey Loucks[28], Jane E. Loveland[23], Jianguo Lu[71], Shuangjia Lu[64,65], Julian K. Lucas[28], Juan F. Macias-Velasco[21,26,72], Kateryna D. Makova[73], Maximillian G. Marin[54,66], Christopher Markovic[21], Tobias Marschall[43,48], Franco L. Marsico[37], Fergal J. Martin[23], Mira Mastoras[28], Capucine Mayoud[50], Brandy McNulty[28], Jack A. Medico[29], Julian M. Menendez[28], Karen H. Miga[28], Anna Minkina[74], Matthew W. Mitchell[75], Saswat K. Mohanty[76], Younes Mokrab[61,77,78], Jean Monlong[79], Shabir Moosa[61], Avelina Moreno-Ochando[80,81], Shinichi Morishita[82], Jonathan M. Mudge[23], Katherine M. Munson[1], Njagi Mwaniki[83], Nasna Nassir[24], Chiara Natali[40], Shloka Negi[28], Lingbin Ni[1], Adam M. Novak[28], Pilar N. Ossorio[84], Chie Owa[82], Sadye Paez[29], Benedict Paten[28], Clelia Peano[31,85], Adam M. Phillippy[27], Brandon D. Pickett[27], Laura Pignata[37], Nadia Pisanti[83], David Porubsky[1,86], Pjotr Prins[37], Anandi Radhakrishnan[28], T. Rhyker Ranallo-Benavidez[30], Brian J. Raney[28], Mikko Rautiainen[87], Alessandro Raveane[31], Luyao Ren[1,20], Arang Rhie[27], Fedor Ryabov[88,89], Samuel Sacco[41], Farnaz Salehi[37], Michael C. Schatz[69,90], Laura B. Scheinfeldt[75], Aarushi Sehgal[52], William E. Seligmann[41], Mahsa Shabani[91], Kishwar Shafin[38], Shadi Shahatit[50], Ruhollah Shemirani[32], Vikram S. Shivakumar[69], Swati Sinha[23], Jouni Sirén[28], Linnéa Smeds[76], Steven J. Solar[27], Marco Sollitto[29,40], Nicole Soranzo[31,46,92], Andrew B. Stergachis[1,74], Marie-Marthe Suner[23], Yoshihiko Suzuki[82], Arda Söylev[43,48], Ahmad Abou Tayoun[93,94], Jack A. S. Tierney[23], Chad Tomlinson[21], Francesca Floriana Tricomi[23], Mohammed Uddin[24,95], Matteo Tommaso Ungaro[41,96], Rahul Varki[33], Flavia Villani[37], Ivo Violich[28], Mitchell R. Vollger[74], Brian P. Walenz[27], Charles Wang[97], Lisa E. Wang[32], Ting Wang[21,26,72], Aaron M. Wenger[98], Conor V. Whelan[29], Zilan Xin[26], Zheng Xu[26], Kai Ye[99], DongAhn Yoo[1], Wenjin Zhang[26], Ying Zhou[54], Xiaoyu Zhuo[26] & Giulia Zunino[31]

[21]McDonnell Genome Institute, Washington University School of Medicine, St. Louis, MO 63108, USA. [22]Department of Human Molecular Genetics and Biochemistry, Faculty of Medical and Health Sciences, Tel Aviv University, Tel Aviv 69978, Israel. [23]European Molecular Biology Laboratory, European Bioinformatics Institute (EMBL-EBI), Wellcome Genome Campus, Hinxton, Cambridge CB10 1SD, UK. [24]Center for Applied and Translational Genomics (CATG), Mohammed Bin Rashid University of Medicine and Health Sciences, Dubai Health, Dubai, UAE. [25]Department of Genetics, Stanford University, Palo Alto, CA 94304, USA. [26]Department of Genetics, Washington University School of Medicine, St. Louis, MO 63110, USA. [27]Genome Informatics Section, Center for Genomics and Data Science Research, National Human Genome Research Institute, National Institutes of Health, Bethesda, MD 20892, USA. [28]UC Santa Cruz Genomics Institute, University of California, Santa Cruz, CA 95060, USA. [29]The Vertebrate Genome Laboratory, The Rockefeller University, New York, NY 10065, USA. [30]Bioinnovation and Genome Sciences, The Translational Genomics Research Institute (TGen), Phoenix, AZ 85004, USA. [31]Human Technopole, Milan, Italy. [32]Institute for Genomic Health, Icahn School of Medicine at Mount Sinai, New York, NY 10029, USA. [33]Department of Computer and Information Science and Engineering, University of Florida, Gainesville, FL 32611, USA. [34]Canadian Center for Computational Genomics, McGill University, Montréal, QC H3A 0G1, Canada. [35]Department of Human Genetics, McGill University, Montréal, QC H3A 0G1, Canada. [36]Victor Phillip Dahdaleh Institute of Genomic Medicine, Montréal, QC H3A 0G1, Canada. [37]Department of Genetics, Genomics and Informatics, University of Tennessee Health Science Center, Memphis, TN 38163, USA. [38]Google LLC, Mountain View, CA 94043, USA. [39]Department of Biomedical Informatics and Data Science, Yale School of Medicine, New Haven, CT, USA. [40]Department of Biology, University of Florence, Sesto Fiorentino, FI 50019, Italy. [41]Department of Ecology and Evolutionary Biology, University of California, Santa Cruz, CA 95060, USA. [42]Arizona State University, Consortium for Science, Policy & Outcomes, Washington, DC 20006, USA. [43]Center for Digital Medicine, Heinrich Heine University Düsseldorf NRW, Düsseldorf, DE, Germany. [44]Department for Endocrinology and Diabetology at the Medical Faculty and University Hospital Düsseldorf, Heinrich Heine University Düsseldorf NRW, Düsseldorf, DE, Germany. [45]Paul-Langerhans-Group Computational Diabetology, German Diabetes Center (DDZ) and Leibniz Institute for Diabetes Research NRW, Düsseldorf, DE, Germany. [46]Wellcome Sanger Institute, Genome Campus, Hinxton CB10 1RQ, UK. [47]Department of Genetics, University of Cambridge, Cambridge CB2 3EH, UK. [48]Institute for Medical Biometry and Bioinformatics, Medical Faculty and University Hospital Düsseldorf, Heinrich Heine University NRW, Düsseldorf, DE, Germany. [49]Howard Hughes Medical Institute, Chevy Chase, MD 20815, USA. [50]ISEM, Univ Montpellier, CNRS, IRD, Montpellier, France. [51]Institut Universitaire de France, Paris, France. [52]Department of Computer Science and Engineering, University of California San Diego, La Jolla, CA 92093, USA. [53]Department of Bioethics & Humanities, University of Washington School of Medicine, Seattle, WA 98195, USA. [54]Department of Data Science, Dana-Farber Cancer Institute, Boston, MA 02215, USA. [55]Department of Anthropology, University of Kansas, Lawrence, KS 66045, USA. [56]School of Health Sciences, University of Manchester, Manchester M13 9PL, UK. [57]Traditional, ancestral and unceded territory of the Gabrielino/Tongva peoples, Institute for Society & Genetics, University of California, Los Angeles, Los Angeles, CA 90095, USA. [58]Traditional, ancestral and unceded territory of the Gabrielino/Tongva peoples, Institute for Precision Health, David Geffen School of Medicine, University of California, Los Angeles, Los Angeles, CA 90095, USA. [59]Traditional, ancestral and unceded territory of the Gabrielino/Tongva peoples, Division of General Internal Medicine & Health Services Research, David Geffen School of Medicine, University of California, Los Angeles, Los Angeles, CA 90095, USA. [60]Department of Electrical Engineering and Computer Science, University of California, Berkeley, Berkeley, CA 94720, USA. [61]Medical and Population Genomics Lab, Sidra Medicine, Doha, Qatar. [62]Montreal Heart Institute, Montréal, QC, Canada. [63]Department of Pediatrics, University of California San Diego, La Jolla, CA 92093, USA. [64]Center for Genomic Health, Yale University School of Medicine, New Haven, CT, USA. [65]Department of Genetics, Yale University School of Medicine, New Haven, CT, USA. [66]Department of Biomedical Informatics, Harvard Medical School, Boston, MA, USA. [67]Department of Pediatrics, Division of Genetics, School of Medicine, University of California, Irvine, CA 92697, USA. [68]Department of Evolution and Ecology and the Center for Population Biology, University of California, One Shields, Davis, CA 95616, USA. [69]Department of Computer Science, Johns Hopkins University, Baltimore, MD, USA. [70]Department of Genetics, Epigenetics Institute, Perelman School of Medicine, University of Pennsylvania, Philadelphia, PA, USA. [71]Sun Yat-sen University, Guangzhou, China. [72]Edison Family Center for Genome Sciences & Systems Biology, Washington University School of Medicine, St. Louis, MO 63110, USA. [73]Department of Biology and Center for Medical Genomics, Penn State University, University Park, PA 16802, USA. [74]Division of Medical Genetics, Department of Medicine, University of Washington School of Medicine, Seattle, WA 98195, USA. [75]Coriell Institute for Medical Research, Camden, NJ 08103, USA. [76]Department of Biology, Penn State University, University Park, PA 16802, USA. [77]Department of Biomedical Science, College of Health Sciences, Qatar University, Doha, Qatar. [78]Department of Genetic Medicine, Weill Cornell Medicine-Qatar, Doha, Qatar. [79]IRSD -

Digestive Health Research Institute, University of Toulouse, INSERM, INRAE, ENVT, UPS, Toulouse, France. [80]MATCH biosystems, S.L, Elche, Spain. [81]Universidad Miguel Hernández de Elche, Elche, Spain. [82]Department of Computational Biology and Medical Sciences, The University of Tokyo, Kashiwa, Chiba 277-8561, Japan. [83]Department of Computer Science, University of Pisa, Pisa, Italy. [84]Law School, University of Wisconsin-Madison, Madison, WI 53706, USA. [85]Institute of Genetics and Biomedical Research, UoS of Milan, National Research Council, Milan, Italy. [86]Genome Biology Unit, European Molecular Biology Laboratory (EMBL), Heidelberg, DE, Germany. [87]Institute for Molecular Medicine Finland, Helsinki Institute of Life Science, University of Helsinki, Helsinki, Finland. [88]The Center for Bio- and Medical Technologies, Moscow, Russia. [89]Centre for Biomedical Research and Technology, HSE University, Moscow, Russia. [90]Department of Biology, Johns Hopkins University, Baltimore, MD, USA. [91]University of Amsterdam, Amsterdam, Netherlands. [92]School of Clinical Medicine, University of Cambridge, Cambridge CB2 0SP, UK. [93]Center for Genomic Discovery, Mohammed Bin Rashid University, Dubai Health, UAE. [94]Dubai Health Genomic Medicine Center, Dubai Health, UAE. [95]GenomeArc Inc, Mississauga, ON, Canada. [96]Department of Biology and Biotechnologies "Charles Darwin", University of Rome "La Sapienza", Piazzale Aldo Moro 00185 RM, Italy. [97]Center for Genomics, Loma Linda University School of Medicine, Loma Linda, CA 92350, USA. [98]PacBio, Menlo Park, CA 94025, USA. [99]The first affiliated hospital of Xi'an Jiaotong University, Xi'an Jiaotong University, Xi'an, Shaanxi 710049, China.

