## [Transparent Peer Review file · Nature Communications]

Using the linear references from the pangenome to discover missing autism variants

Corresponding Author: Professor Evan Eichler

Version 0:

Reviewer comments:

Reviewer #1

(Remarks to the Author)

Sui et al. present a study using long read sequencing (LRS) data from 51 families of unsolved autism cases to perform deep characterization of de novo mutations, structural variants, and methylation. They use newly available LRS/assemblies from controls from HPRC/HGSVC to substantially decrease the search space and identify potential rare/denovo pathogenic variants at structural variants (SVs). Although the study did not find an overall increased SV burden in probands, this is likely due to insufficient power at this sample size.

This study is a valuable proof of principle in showing the promise of using LRS as a diagnostic tool. A major challenge in moving to LRS is a lack of large population control databases of SVs generated by these technologies (compared to up to at least hundreds of thousands of short read datasets that are used as controls for SNVs). Although the control sample set used here is still pretty small, it enables filtering 97% of variants (99% with the expanded control set) which is promising. It is also encouraging that the study found known things (e.g. SYNGAP1, MECP2, TBL1XR1 mutations) that were not found or filtered by short reads. Finally, the study generates high quality LRS data and assemblies that can presumably be used as a resource in future studies.

Overall the paper is well written and easy to follow. I have several comments regarding clarifying some of the terminology used and toning down some of the claims that are made or strengthening them with some additional analysis:

1. The manuscript oversells claims about the completeness of the control variant callsets. In particular, if I understood the sample size correctly (108 controls), most of the variants in controls are probably common and many ultra-rare things are missing. It would be helpful to provide an estimate of the % of variants in the population at different MAF thresholds that are estimated to be in the control set (or some other sort of metrics that could quantify the completeness of the reference. Fig. 5a shows the cumulative SV number but doesn't assess how low of MAF we think that reference captures).

Some specific places in the text that are relevant:

Line 130: The claim that variant discovery from HPRC/HGSVC is complete is misleading. This is true in the sense that all of the variants in each genome are theoretically discovered. But not true in the sense that these resources still have limited sample size and thus are missing many rare variants. This nuance should be clarified.

Line 144: what MAF threshold is being used to define ultra-rare? The "Variant discovery" section of the results mentions 108 population controls (line 189). Which would mean the lowest frequency that can be detected is >0.1%, and likely a large number of rare things are missing from that reference by chance.

In the last section of the paper, the control pool is expanded to 569 individuals. It wasn't clear why not to just use that larger control set in the first place?

2. How much improvement do the LRS control datasets give for filtering control SVs compared to other datasets based on short reads? The authors state that 97% of variants can be filtered using the population controls, which is exciting. Are there available short read SV catalogs that could have been used and would those add anything beyond the LRS controls? It is clear that per genome the number of SVs discovered is far smaller than that from LRS, but the available sample sizes are

much larger which could help with rare variants.

3. I find the usage of the word pangenome here and elsewhere to be confusing. This is largely because the use of the term in general in the field (not specifically in this paper) is unclear. Technically, a pangenome is a collection of genomes from individuals of the same species, which this paper uses. But the word has been commonly associated specifically with using that collection as a reference (rather than a single linear reference) and also to graph-based representations (e.g. as in minigraph, pggg, etc although other paradigms to represent pangenomes exist). This paper does neither of these, and is rather using SVs from these genomes as a set of population controls. If those control SVs had been derived from short reads, e.g. from 1000 Genomes, would that also be considered a pangenome analysis? The title, "Pangenome discovery of missing autism variants", gave me the impression the pangenome would actually be used for genotyping e.g. as a reference rather than mostly for downstream filtering, but it appears genotyping was still done using GRCh38 as a reference. I would suggest revising the title to be less misleading about that and also clarifying in the intro what constitutes pangenome analysis.

Additional points:

The X chromosome result of a trend toward increased SV burden on chrX among affected females is highlighted in the abstract but is not actually statistically significant. While the result is promising, it should be clarified in the abstract that this is only a suggestive trend.

Line 207: The average Mendelian Inheritance rate for SVs across the trios is 90.4%. How does this compare with other SV studies (seems maybe comparable to Liao et al Nature 2023 HPRC paper?). I assume the mutation rate of SVs is far less than 10% so most of the inconsistencies are driven by genotyping errors. What are the implications of that for the potential for false positives here?

Line 255: "The brain-derived regulatory regions (brainREG) annotation resulted in a 31% increase in the total number of regulatory SVs" -> compared to what as a baseline?

Line 250 section "SV burden analysis": This paragraph starts off about regulatory annotations, but I think the second half of the paragraph is focusing on overall burden (starting on line 263). It would be good to clarify which results are overall vs. across the categories tested. If line 263 onwards is not about the regulatory annotations that might be better as a new paragraph.

Line 341: MPRA is not defined

Line 347: worth noting that HNRNPK is also an RNA binding protein

(Remarks on code availability)

Reviewer #2

(Remarks to the Author)

The manuscript submitted by Evan Eichler, Huda Zoghbi and colleagues is first and foremost a novel conceptual paper in the implementation of near-complete T2T sequencing into the clinic. In this way it sets standards and guides future T2T diagnostic papers.

The authors analysed by near-complete genome sequencing 189 individuals from 51 families with unsolved autism.

Specifically the authors claim that most of the variants identified could not have been detected by srGS.

Bioinformatic tools were more or less the today's standard tools in the field showing, that wetlab technology and bioinformatics are ready to enter the clinic. There will be further improvements over time as the current GRCh38 reference data set still has numerous gaps or errors, however, this manuscript adds to additional samples to be used as reference data sets.

The authors state nicely why srGS might have failed to detect several clear causative variants. However, it seems to me that standard long read GS may have detected these variants as well and a de novo assembly might not have been necessary, please comment on this.

A couple of potential novel findings have been made like identification of repeat expansions exclusively in males inherited from their mothers and several patients with skewing of X inactivation. As there are not suppl data on clinical presentation and as the cohort most likely is quite heterogeneous (already based on the molecular identification of Rett syndrome in the cohort) it may add to the manuscript to mention whether there are any specifics on these patients (such as developmental issues, IQ or severe forms of autism), please add.

Finally, in several of the potentially identified mutations (Table 1) novel regulatory mutations may have been identified. Many of them are easily detected by RNAseq analysis. It would clearly add to the manuscript to have these verified whether or not the variants led to a reduced gene expression.

Minor comments:

page 4, line 102 "conventional SRS approaches such as Illumina whole-genome sequencing..."

I guess this would have been similar with BGI or Ultima (or other srGS) based technologies, so I would delete the word

"Illumina"

page 4, line 119.. it seems to me that the references do not fit entirely

page 5, line 141 ...in the proband via conventional Illumina SRS... please specify, do you mean Illumina whole genome SRS? or was this to a certain percentage also WES?

page 5, line 168, I did not understand why you had to use parental Illumina reads for hifiasm to generate the parental haplotypes as you had already each person sequenced with PacBio.

page 10, lines 281 - 285, the authors identified three hemizygous repeat outliers in males, all inherited from the mother. Based on the limited reference data today, it can not be judged whether or not these are causal repeat expansions. In this chapter the authors refer only to the IL1RAPL1 but not to the F9 and intergenic repeat expansion (all provided correctly in suppl S8), however for easier readability it would be nice to list them here all. Any specifics on the country of origin?

page 10, line 285: I guess between "....SFARI score 2 gene). andBecause LRS data allow..." would be a new paragraph

Olaf Riess

(Remarks on code availability)

I'm not a bioinformatician thus I did not download or use the code.

Reviewer #3

(Remarks to the Author)

The genetic basis of autism remains underexplored. In this study, the authors performed genome sequencing of 189 individuals from 51 families with unsolved cases and generated a set of high-quality assemblies. These assemblies were compared with publicly available high-quality assemblies from healthy donors, leading to the discovery of three pathogenic variants in TBL1XR1, MECP2, and SYNGAP1, as well as nine candidate variants, most of which were missed by short-read sequencing. The study is well designed and solid. I have a few concerns as outlined below.

Major comments

MECP2 is structurally complex with diverse SVs and CNV events. I am wondering whether this case also harbors any other events besides the discovered deletion. Moreover, what is the actual deleted sequence in the last exon region, and what is the precise effect of this deletion on the last exon?

The paper includes a large control cohort, which is very good. However, controls appear underrepresented for EUR ancestry (e.g., 39 EUR among 569). Please report the ancestry and sex composition of controls vs. families, and perform ancestry- and sex-matched analyses.

In Fig. 1, the authors showed the N50 of contigs in autism family assemblies. However, providing NGx curves against GRCh38 and CHM13 would still be valuable. Comparing family assemblies with control assemblies would also be helpful.

The authors claim a "modest rate of pathogenic variant discovery (5.9%)." Did the authors consider evaluating compound heterozygous and oligogenic models (e.g., small variants, or small variants + SVs)? This may help recover additional pathogenic candidates.

Please describe unpublished tools (e.g., subseq, BoostSV) and provide versions, key parameters, and workflow availability to enhance reproducibility.

Minor comments

Fig. 2a: Distinguish proband vs. sibling more clearly (shapes or colors).

Fig. 4c,d: Correct "Genehance" → "GeneHancer."

Typo: "TBXLR1" → "TBL1XR1."

I would like to thank the authors for their contribution to the field through this work.

(Remarks on code availability)

I would appreciate for providing the documents for some unpublished tools.

Version 1:

Reviewer comments:

Reviewer #1

(Remarks to the Author)

The authors have thoroughly addressed all of my comments. The manuscript has improved substantially and is an important

contribution to the field.

(Remarks on code availability)

The code and documentation for the analysis is very detailed and the authors have done a great job.

I noticed many links say "Internal path" and link to an Eichler Lab site which requires login info, and therefore is not publicly accessible or fully reproducible.

Reviewer #2

(Remarks to the Author)

(Remarks on code availability)

Reviewer #3

(Remarks to the Author)

I have carefully reviewed the authors' revisions, and I think all of my previous concerns have been fully addressed. I have no further comments at this time.

(Remarks on code availability)

Source code has been properly deposited in several public repositories. I thank the authors for their efforts.

Reviewer #1 (Remarks to the Author):

Sui et al. present a study using long read sequencing (LRS) data from 51 families of unsolved autism cases to perform deep characterization of de novo mutations, structural variants, and methylation. They use newly available LRS/assemblies from controls from HPRC/HGSVC to substantially decrease the search space and identify potential rare/denovo pathogenic variants at structural variants (SVs). Although the study did not find an overall increased SV burden in probands, this is likely due to insufficient power at this sample size.

This study is a valuable proof of principle in showing the promise of using LRS as a diagnostic tool. A major challenge in moving to LRS is a lack of large population control databases of SVs generated by these technologies (compared to up to at least hundreds of thousands of short read datasets that are used as controls for SNVs). Although the control sample set used here is still pretty small, it enables filtering 97% of variants (99% with the expanded control set) which is promising. It is also encouraging that the study found known things (e.g. SYNGAP1, MECP2, TBL1XR1 mutations) that were not found or filtered by short reads. Finally, the study generates high quality LRS data and assemblies that can presumably be used as a resource in future studies.

Overall the paper is well written and easy to follow. I have several comments regarding clarifying some of the terminology used and toning down some of the claims that are made or strengthening them with some additional analysis:

We are grateful to the reviewer for their careful reading and valuable suggestions.

1. The manuscript oversells claims about the completeness of the control variant callsets. In particular, if I understood the sample size correctly (108 controls), most of the variants in controls are probably common and many ultra-rare things are missing. It would be helpful to provide an estimate of the % of variants in the population at different MAF thresholds that are estimated to be in the control set (or some other sort of metrics that could quantify the completeness of the reference. Fig. 5a shows the cumulative SV number but doesn't assess how low of MAF we think that reference captures).

This is a good suggestion. We added some comments to the text regarding the "completeness" of the 108-control set and the possibility of missing variants based on sample size. To address the comment of how many SVs are likely missing in that limited set, we created a population SV set and plotted the MAF in the 108-control set to

the larger sample set. From that, we computed the number of missing variants in the 108 controls as a function of MAF in 569 controls and 102 unrelated parents (n=1,342).

We added the following statement and one SI figure (with underline indicating the modifications in response to Reviewer #3):

“Increasing the number of controls, especially samples of African origin, nearly doubles the number of SVs from 271,375 (108 controls) to 445,142 (569 controls) nonredundant SVs (Supplementary Data 5-8). **We generated a population-level SV reference and compared the MAF distribution between the 108 controls and the largest dataset comprising 569 controls and 102 unrelated parents (n = 1,342). Based on this comparison, we estimated the fraction of variants captured at different MAF thresholds (Supplementary Fig. 12). Notably, only 16.2% of SVs with MAF < 0.1% in the larger dataset were detected in the 108-control set, illustrating the limited sensitivity of the smaller reference for identifying ultra-rare variants.** In an ancestry- and sex-matched analysis restricted to individuals of European ancestry (38 autism families and 73 controls), each child carried an average of 754 rare SVs. Focusing on rare variants absent from the largest 569-control set, the number of rare SVs concomitantly drops to 202 events per sample (Fig. 5b), corresponding to ~74% of rare variants being filtered out by the more diverse population controls. This demonstrates the substantial gain in sensitivity for removing common and low-frequency variants when using a larger, heterogeneous reference panel. As a result, 99% of common SVs are excluded per individual forming a more tractable and potentially biomedically relevant set of rare variants for downstream interpretation and enrichment analyses....”

Supplementary Figure 12. Effect of control cohort size on SV discovery and minor allele frequency (MAF) spectrum. The plot shows the fraction of SVs captured (blue) in an initial control set of 108 genomes across different MAF thresholds derived from an expanded dataset comprising 569 controls and 102 unrelated parents (1,342 haplotypes in total). Only 16.2% of SVs with MAF < 0.1% in the larger dataset were detected in the smaller 108-control reference, underscoring the limited sensitivity of the smaller cohort for identifying ultra-rare variants.

Some specific places in the text that are relevant:

Line 130: The claim that variant discovery from HPRC/HGSVC is complete is misleading. This is true in the sense that all of the variants in each genome are theoretically discovered. But not true in the sense that these resources still have limited sample size and thus are missing many rare variants. This nuance should be clarified.

We revised the sentence to reflect this nuance as suggested:

“This resource is potentially valuable to the clinical genetics community because variant discovery is **more** complete, providing a control set to assess the frequency of variants in regions typically unassayable by SRS and therefore absent or unreliable in associated databases such as gnomAD (Collins et al. 2020; S. Chen et al. 2024). Moreover, assembly-based comparisons between offspring and parental genomes have been shown to further increase the power to discover DNM by essentially eliminating reference biases (Porubsky et al, 2025). **Notwithstanding, the number of samples still remains modest and much larger sample sizes will be required to understand the full spectrum especially for those with lower minor allele frequency (MAF).**”

Line 144: what MAF threshold is being used to define ultra-rare? The “Variant discovery” section of the results mentions 108 population controls (line 189). Which would mean the lowest frequency that can be detected is >0.1%, and likely a large number of rare things are missing from that reference by chance.

That is a good catch as they are hardly ultra-rare at a frequency of <1/216 from 108 population controls. We revised the sentence accordingly:

“...we then compared with HPRC and HGSVC population controls to discover and validate *de novo* and **rare variants (<0.5%)** as new candidates for autism.”

In the last section of the paper, the control pool is expanded to 569 individuals. It wasn't clear why not to just use that larger control set in the first place?

Our experiments were staged over a year and reflect a continued expansion of the 1000 Genomes Project (1KGP) samples as part of ongoing efforts from our laboratory and others. We felt that showing this progression might be useful to the research community as the controls used have diverse ancestry. As described both in the first and last Results sections, we used the initial control set of 108 individuals from the HPRC and HGVC, previously published in Liao et al. 2023, Logsdon, Rozanski, et al. 2024 and Ebert et al. 2021. We then analyzed 177 newly released HPRC samples ourselves, expanding the set to 285. To further increase the power, we subsequently included 284 additional ONT controls from the 1KGP, generating the larger set of 569 controls. These stepwise expansions reflect the availability of data at the time of analysis, while also highlighting that larger control sets substantially increase statistical power and robustness, allowing us to better distinguish rare signals from background common variants as now nicely illustrated by the suggested Supplementary Fig. 12. To clarify, we now include citations of published controls and revised the text to reflect this aspect of the experimental design:

“We aggregated the validated SVs from all 189 study samples and compared them with SVs observed in the 108 population controls from the HPRC and HGVC (Ebert et al. 2021; Logsdon, Rozanski, et al. 2024; Liao et al. 2023) via Truvari (English et al. 2022), focusing on rare SVs exclusive to autism families. **As a result of subsequent LRS of additional 1000 Genomes Project (1KGP) samples recently released as part of the HPRC and other efforts (Gustafson et al. 2024), we expanded our SV callset from controls in a staged manner, allowing us to access variants of reduced MAF (see below Supplementary Fig. 12).**”

2. How much improvement do the LRS control datasets give for filtering control SVs compared to other datasets based on short reads? The authors state that 97% of variants can be filtered using the population controls, which is exciting. Are there available short read SV catalogs that could have been used and would those add anything beyond the LRS controls? It is clear that per genome the number of SVs discovered is far smaller than that from LRS, but the available sample sizes are much larger which could help with rare variants.

We performed the suggested comparison using GATK_SV calls from 63,000 genomes from gnomAD SRS data and the filtering gain was marginal (0.27%, 55/20,716 SVs). By contrast, if we limited to SRS, a large fraction of SVs (~68%, 14,110/20,716) remained undetected. Because we believe this analysis is of interest to the general readership, we added the following paragraph (in bold) to the last result section of the main text:

“To enhance the power of pangenome filtering to effectively exclude common SVs and focus on a high-confidence pool of rare variants, we expanded the control cohort size from 108 to 285 and then to 569 individuals.

We also compared our LRS SV datasets with a comprehensive SRS SV callset (INS and DEL) generated by GATK-SV (Collins et al. 2020) from 63,046 unrelated genomes, as represented in gnomADv4.1 (Collins et al. 2020). To ensure a fair comparison between SRS and LRS datasets that differ in resolution and breakpoint precision, we used relaxed matching criteria in Truvari bench: a minimum of 50% reciprocal overlap, 50% size similarity, maximum breakpoint distance of 500 bp, and no sequence similarity requirement. We added the gnomAD SV IDs to the supplementary datasets (Supplementary Data 2, 5 and 7, Supplementary Fig. 14). Within the 108-control set, 162,916 of 271,375 total nonredundant SVs were observed in the autism families. Of these, 77.6% (126,377 SVs) were present in the 108 LRS controls, yielding 663 rare SVs per child. In contrast, only 24.1% (39,234 SVs) overlapped with the SRS-based gnomAD SVs from 63,046 unrelated controls (supplementary Fig. 14), corresponding to 14,110 rare SVs per child when considering only short-read controls. Thus, despite its much larger sample size, the short-read dataset can be considered an auxiliary resource for further filtering rare SVs beyond those already excluded by the LRS controls, underscoring the substantially greater power of LRS data for rare variant filtering. Specifically, incorporating the SRS dataset would have further excluded 0.71% (147/20,716 per child) and 0.27% (55/20,716 per child) of SVs from the 108- and 569-control sets, respectively. Given the inherent differences in breakpoint resolution and variant representation between LRS and SRS platforms, these overlaps reflect potentially inflated estimates. A reassuring sign of the stringency and completeness of our filtering strategy is that all 12 candidates identified were rare variants absent from both the LRS and SRS control datasets.”

Also, in the Methods:

“SV merging and filtering strategy. The strategy involved **three** major steps. Briefly,

1. Callerset validation. ...
2. Inter-sample merge. ...
3. Rare SV pool discovery. ...

Separately, we compared our LRS SV datasets with the SRS control set from 63,046 unrelated genomes in gnomAD v4.1 to evaluate the overlap between platforms. SV comparison was conducted using Truvari bench with relaxed

matching parameters to account for differences in breakpoint precision between LRS and SRS datasets:

```
truvari bench -c {gnomad.v4.1.INSDEL50.non_neuro_controls.sites.vcf.gz} -b {108/285/569ctr_189asd_collapsed.vcf.gz} --pctsize 0.5 --pctseq 0.0 --pctovl 0.5 --sizefilt 50 --sizemax 100000 -o {output}
```

We incorporated gnomAD SV IDs, overlap metrics (PctSizeSimilarity:PctRecOverlap:SizeDiff), and allele counts (heterozygous, homozygous, and sex-specific; overall and within the non-neuro subset) into the collapsed SV tables (Supplementary Data 2, 5 and 7). Similar to step 3 above, we applied sex- and zygosity-aware filtering to identify rare SVs absent from all SRS genomes.

Supplementary Figure 14. Comparison of nonredundant SVs observed in LRS and SRS control sets. Collapsed nonredundant SVs from the three LRS control datasets were compared against SRS-based SVs (INS and DEL) generated by GATK-SV from 63,046 unrelated genomes in gnomAD v4.1 (corresponding to Supplementary Data 2, 5 and 7). Overlap between LRS and SRS SVs was defined using relaxed matching criteria in Truvari bench: a minimum of 50% reciprocal overlap, 50% size similarity, and maximum breakpoint distance of 500 bp, with no sequence similarity requirement. Cross-hatched bars indicate SVs that would have been detected by SRS.

3. I find the usage of the word pangenome here and elsewhere to be confusing. This is largely because the use of the term in general in the field (not specifically in this paper) is unclear. Technically, a pangenome is a collection of genomes from individuals of the same species, which this paper uses. But the word has been commonly associated specifically with using that collection as a reference (rather than a single linear reference) and also to graph-based representations (e.g. as in minigraph, pggg, etc although other paradigms to represent pangenomes exist). This paper does neither of these, and is rather using SVs from these genomes as a set of population controls. If those control SVs had been derived from short reads, e.g. from 1000 Genomes, would that also be considered a pangenome analysis? The title, "Pangenome discovery of missing autism variants", gave me the impression the pangenome would actually be used for genotyping e.g. as a reference rather than mostly for downstream filtering, but it appears genotyping was still done using GRCh38 as a reference. I would suggest revising the title to be less misleading about that and also clarifying in the intro what constitutes pangenome analysis.

Yes, there does seem to be confusion in the field in part because terms need to be clearly defined. There are three definitions/activities that are distinguishable: 1) the generation of the pangenome, which as indicated by the referee is simply the collection of nearly complete linear references; 2) construction of a graph-based pangenome, which combines these linear references to a singular reference where variants are captured and correctly represented by the underlying graph structure; and 3) the use of that graph-based reference or the collection of linear references themselves for (hopefully improved) genotyping into whole-genome sequencing data. Our paper focuses on #1 and #3 without the use of a graph, i.e., we only use the collection of linear references. While it is true that SRS datasets from many human genomes might technically be considered a pangenome if SRS were assembled, implicit in a pangenome design is the construction of an *ab initio* assembly first, prior to variant discovery as opposed to aligning reads to a singular reference, which is how SVs are typically discovered by SRS and would not qualify as a pangenomic approach.

While we prefer the shorter title and believe it is valid, a more descriptive title that would make it clear that we are not employing a graph could be:

"Using the linear references from the pangenome to discover missing autism variants"

Additional points:

The X chromosome result of a trend toward increased SV burden on chrX among affected females is highlighted in the abstract but is not actually statistically significant. While the result is promising, it should be clarified in the abstract that this is only a suggestive trend.

We revised the abstract to clarify that the observation represents a suggestive trend rather than a statistically significant result. The revised abstract now reads:

"...yet **observe a suggestive trend toward an increased SV burden** on the X chromosome among affected females."

We also made this clearer in the legend:

"No significant differences (χ^2 test p-values exceeding 0.05) in the number of SVs between probands and siblings were observed across these categories, with a **still insignificant** trend observed on the X chromosome for enrichment of SVs on affected females compared to unaffected sisters."

Line 207: The average Mendelian Inheritance rate for SVs across the trios is 90.4%. How does this compare with other SV studies (seems maybe comparable to Liao et al Nature 2023 HPRC paper?). I assume the mutation rate of SVs is far less than 10% so most of the inconsistencies are driven by genotyping errors. What are the implications of that for the potential for false positives here?

Our Mendelian inheritance rate is comparable to other studies that have investigated using LRS and called SVs (Duan et al., *BMC Genomics*, 2002; Schloissnig et al., *Nature*, 2025). There are at least two factors: the individual callers used and the type of variants considered. Mendelian errors are highest for multi-allelic variants (e.g., VNTRs) where there have been reports as high as 15%. It should be noted however that such variants are particularly prone to alignment artefacts as well as sequencing artefacts—so there is also a reproducibility in calling. We have expanded upon this to make the implications with respect to the nature of the false positives clearer:

"The average Mendelian concordance rate for SVs across 87 trios is 90.4% **with many of the discrepant alleles associated with multiallelic variants, such as variable number tandem repeats (VNTRs) where both sequence and alignment artefacts confound variant calling (Schloissnig et al. 2025).**"

Line 255: "The brain-derived regulatory regions (brainREG) annotation resulted in a 31% increase in the total number of regulatory SVs" -> compared to what as a baseline?

We apologize for this confusing sentence and the 31% statistic was not correct. The additional annotation provided by ATAC-seq and CUT&Tag experiments allowed us to identify an additional 6,171 SVs that intersect with brain-specific regulatory regions that were not defined by ENCODE.

"The additional annotation allowed us to identify another 6,171 SVs associated with brain-derived regulatory regions (brainREG) beyond those defined by ENCODE intersection and consequently a 45% (6,171/13,773) increase in potential SVs affecting noncoding regulatory DNA (Fig. 2c-d)."

Line 250 section "SV burden analysis": This paragraph starts off about regulatory annotations, but I think the second half of the paragraph is focusing on overall burden (starting on line 263). It would be good to clarify which results are overall vs. across the categories tested. If line 263 onwards is not about the regulatory annotations that might be better as a new paragraph.

This is a good suggestion so we split the paragraph as recommended:

"... The rates of autosomal SVs (Fig. 2c, Supplementary Fig. 6) do not differ significantly between probands and unaffected siblings across various categories (nominal $p > 0.05$, OR < 1 , χ^2 test).

For X chromosome SVs, we analyzed males and females separately. ..."

Line 341: MPRA is not defined

We now define the MPRA acronym as "**massively parallel reporter assay**" in the text.

Line 347: worth noting that HNRNPK is also an RNA binding protein

We revised the text to indicate that HNRNPK, in addition to acting as a transcription factor, also functions as an RNA-binding protein:

"This INS is predicted to disrupt HNRNPK transcription factor (TF) binding sites in 11201_p1; **notably, HNRNPK also functions as an RNA-binding protein.**"

Reviewer #2 (Remarks to the Author):

The manuscript submitted by Evan Eichler, Huda Zoghbi and colleagues is first and foremost a novel conceptual paper in the implementation of near-complete T2T sequencing into the clinic. In this way it sets standards and guides future T2T diagnostic papers.

The authors analysed by near-complete genome sequencing 189 individuals from 51 families with unsolved autism. Specifically the authors claim that most of the variants identified could not have been detected by srGS.

Bioinformatic tools were more or less the today's standard tools in the field showing, that wetlab technology and bioinformatics are ready to enter the clinic. There will be further improvements over time as the current GRCh38 reference data set still has numerous gaps or errors, however, this manuscript adds to additional samples to be used as reference data sets.

We thank the referee for their constructive and helpful comments.

The authors state nicely why srGS might have failed to detect several clear causative variants. However, it seems to me that standard long read GS may have detected these variants as well and a de novo assembly might not have been necessary, please comment on this.

The referee raises a good point. While we found local assembly helpful for confirming parent-of-origin, the missing SRS variants did not, in most cases, require an assembly. We added that point to the first paragraph of the discussion:

“Our analysis of 51 families mostly with daughters affected with autism (often more severely) where we attempted to sequence and assemble the entire euchromatic portion of each genome suggests a more modest rate of pathogenic variant discovery (5.9%). We consider this yield of new pathogenic variant discovery low given that LRS application increased genome-wide sensitivity of DNM detection by 20-40% (Noyes et al. 2025) and we purposefully sequence affected females where the probability of discovery of a large effect mutation is expected to be higher (Lossifov et al. 2014). **It should be noted that LRS alignment to a reference was sufficient to detect these variants previously missed by SRS. Genome assemblies, while helpful for parent-of-origin determination and phasing variants, were not critical to the discovery of most variants.**”

A couple of potential novel findings have been made like identification of repeat expansions exclusively in males inherited from their mothers and several patients with

skewing of X inactivation. As there are not suppl data on clinical presentation and as the cohort most likely is quite heterogeneous (already based on the molecular identification of Rett syndrome in the cohort) it may add to the manuscript to mention whether there are any specifics on these patients (such as developmental issues, IQ or severe forms of autism), please add.

Thank you for pointing this out. We further added available Seizure and Calibrated Severity Score (CSS) from the ADOS to Supplementary Data 1 and evaluated the IQ, CSS and Seizure features in the patients with potential novel findings.

We revised the text as follows:

“The analysis highlighted three hemizygous tandem repeat (TR) expansion outliers in male probands (Supplementary Fig. 9, Methods), each inherited from the mother **in the intron of *IL1RAPL1* (Interleukin 1 Receptor Accessory Protein Like 1, SFARI score 2 gene), the intron of *F9* (Coagulation Factor IX), or the intergenic region between *MXRA5* (Matrix Remodeling Associated 5) and *SNORA48B* (Small Nucleolar RNA, H/ACA Box 48B)**. These longer TR noncoding variants have only been observed in heterozygous states in females and variants of such lengths have yet to be observed in controls. **Notably, the corresponding probands exhibited features commonly associated with neurodevelopmental delay (Supplementary Fig. 9): IQ scores of 18, 13, and 20, and calibrated severity scores (CSS) of 9, 6, with the CSS data for the third not available.**”

Supplementary Figure 9. Outlier TR expansions transmitted from mothers to male probands. **a)** Allele lengths (generated by TRGT) of three TR catalogs (Porubsky, Dashnow, et al. 2025) are shown in gray for unaffected individuals (285 controls and 138 unaffected samples) and in blue for affected probands. The three outlier expansions are highlighted in red and dark gray. **b)** Transmission patterns from maternal haplotypes to male probands are illustrated using multiple sequence alignments (MSAs), with comparisons to the GRCh38 reference. Note that females, with two X chromosomes, display bidirectional TR lengths (both alleles), whereas males, with only one X chromosome, show unidirectional TR length from a single allele. **c)** IQ and calibrated severity scores (CSS) of probands are shown (corresponding to Supplementary Data 1). Values for the three samples carrying outlier TR expansions on the X chromosome are indicated with red lines. Missing information denotes unavailable data.

Finally, in several of the potentially identified mutations (Table 1) novel regulatory mutations may have been identified. Many of them are easily detected by RNAseq

analysis. It would clearly add to the manuscript to have these verified whether or not the variants led to a reduced gene expression.

This was a very interesting suggestion. We attempted to address this comment by performing RNA-seq as well as Kinnex (long-read transcriptomic) sequencing on homozygous deletions (n=5) and promoter substitutions (n=2) where the associated genes would be expressed in the lymphoblastoid cell line. This limited the analysis to seven families, and we did not observe significantly reduced expression for these candidates, probably because they were mainly derived from blood samples.

We added these additional experimental details to the results and methods section:

Results section: *Pathogenic and autism candidate variant discovery.*

“To assess the potential impact of candidate variants, we performed differential expression analysis using long-read RNA sequencing of lymphoblastoid cell lines generated with the PacBio Kinnex platform. Gene-level counts were obtained with IsoQuant (Prjibelski et al. 2023) and analyzed with DESeq2 (Love et al. 2014) for probands carrying genes with two promoter mutations and five homozygous deletions that were expressed in blood. Across all seven candidate loci, no significant reduction in gene expression was detected in blood-derived RNA samples from the target proband compared to the remaining samples (Supplementary Fig. 11). The absence of detectable expression changes may reflect tissue-specific regulatory effects, as these variants are more likely to exert functional consequences in brain tissues, which are directly relevant to the disorder etiology.”

Methods section:

“Long-read Kinnex generation and analysis. Total RNA was extracted from 5 million lymphoblast cells using the RNeasy mini kit from Qiagen. After initial QC using DS-11 FX (Denovix) for UV-Vis quantification and Bioanalyzer RNA 6000 Nano (Agilent, P/N G2939A and 5067-1511) for quality score calculation, 150-250 ng per sample were processed into cDNA and sequencing library concatamers using the Iso-Seq Express 2.0 and Kinnex full-length RNA kits (PacBio, P/N 103-071-500 and 103-072-000) according to manufacturer’s protocols with the following modifications: amplified cDNAs were analyzed for average size using Femto Pulse (Agilent P/N M5330AA and FP-1003-0275) and quantified using fluorescence (Qubit HS DNA Fisher Scientific P/N Q32854) before equimolar pooling and an additional 1.1X SMRTbell Cleanup Bead wash to remove residual primer-dimer before proceeding to Kinnex PCR. Kinnex concatamers were checked for length using Femto Pulse (P/N FP-1002-0275) before loading on 1

SMRT Cell 25M on the PacBio Revio sequencing platform using SPRQ chemistry with Adaptive Loading and a 30-hour acquisition time. After sequencing, raw data were processed with the Read Segmentation and Iso-Seq workflow in SMRT Link v25.3 using default parameters to generate sample-demultiplexed full-length reads (finc.bam) files.

BAM files were converted to FASTA format using SAMtools (v1.21; Danecek et al. 2021) and aligned to the GRCh38 reference genome with minimap2 (v2.28; Li 2021) using the parameters -ax splice:hq -uf --secondary=no --eqx -K 200M. Gene and transcript quantifications were performed with IsoQuant (v3.10.0; Prjibelski et al. 2023) using GENCODE v49 annotations. Raw gene counts from the target proband and remaining samples were used as case and control groups, respectively, for differential expression analysis with DESeq2 (v1.50.0; Love et al. 2014), incorporating sex and diagnosis as variables.”

Supplementary Figure 11. Blood-derived long-read RNA sequencing (PacBio Kinnex) of seven candidate genes. Normalized read counts were generated using the counts() function from DESeq2 for 11 samples (including seven target probands and randomly selected probands and siblings from families 11201 and 14232). Adjusted p-values were calculated by comparing each target proband to the remaining samples using the Wald test in DESeq2.

Minor comments:

page 4, line 102 "conventional SRS approaches such as Illumina whole-genome sequencing..."

I guess this would have been similar with BGI or Ultima (or other srGS) based technologies, so I would deleted the word "Illumina"

We agree that the statement applies broadly to short-read genome sequencing technologies. We revised the text as suggested removing the reference to Illumina:

"... using conventional SRS approaches **associated with** whole-genome sequencing..."

page 4, line 119.. it seems to me that the references do not fit entirely

The cited studies are intended to support the general statement that long-read sequencing (LRS) is increasingly applied to unsolved patients and disorders, in addition to the specific examples mentioned. We revised the sentence to make this clearer:

"Consequently, LRS has been increasingly applied to a variety of unsolved patients and disorders to enhance pathogenic variant discovery, although most studies to date have involved relatively modest cohorts focused almost entirely on read-based discovery (**Hiatt et al. 2021, 2024; Sanchis-Juan et al. 2023; Pauper et al. 2021; Sinha et al. 2025; Steyaert et al. 2025**). For example, Hiatt et al. reanalyzed 10 NDD families and 86 probands using PacBio high-fidelity (HiFi) LRS and found an additional yield of 7.3% beyond SRS mainly in coding regions (**Hiatt et al. 2021; 2024**). Sanchis-Juan et al. applied ONT LRS to complement SRS in four probands (**Sanchis-Juan et al. 2023**)."

page 5, line 141 ...in the proband via conventional Illumina SRS... please specify, do you mean Illumina whole genome SRS? or was this to a certain percentage also WES?

For 46 of the families it was WGS but for the 5 Rett-like samples it involved WES. As described in detail in the Methods section, the unsolved probands were screened using Illumina WGS (n=46, SSC and SAGE cohorts) as well as WES or targeted gene panel testing (n=5, Rett cohort). We revised the text to:

"In this study, we present our initial LRS and assembly resource of 189 individuals from 51 unsolved autism families where no pathogenic variant was previously identified in the proband via conventional Illumina **whole-genome, whole-exome, or gene panel testing with SRS (Methods)**."

page 5, line 168, I did not understand why you had to use parental Illumina reads for hifiasm to generate the parental haplotypes as you had already each person sequenced with PacBio.

Although all individuals were sequenced with PacBio HiFi, trio-binning in hifiasm requires parental Illumina WGS data to provide reliable parental-specific k-mers for phasing. These k-mers are then used to assign the child's HiFi reads to maternal (hap2) and paternal (hap1) haplotypes (also called phased or haplotype-resolved assemblies). Importantly, by leveraging the family pedigree, we are able to generate more accurate and complete phased genomes for the children, making it straightforward to determine the parental origin of variants — a key strength of our study.

To clarify, we revised the sentence to:

"Using parental Illumina reads and hifiasm (Cheng et al. 2021), we generated haplotype-resolved genome assemblies (**i.e., phased by parent-of-origin**) for each of the 87 offspring."

These details were also described in the Methods section:

"Phased genome assembly construction. The HiFi assembly was constructed by hifiasm (v0.16.1, Cheng et al. 2024). Parental short reads were processed with yak (v0.1, <https://github.com/lh3/yak.git>) and then hifiasm trio-binning mode was used for phasing children samples, while the parental assemblies were partially phased by default."

page 10, lines 281 - 285, the authors identified three hemizygous repeat outliers in males, all inherited from the mother. Based on the limited reference data today, it can not be judged whether or not these are causal repeat expansions. In this chapter the authors refer only to the IL1RAPL1 but not to the F9 and intergenic repeat expansion (all provided correctly in suppl S8), however for easier readability it would be nice to list them here all. Any specifics on the country of origin?

We agree with the reviewer and revised the text accordingly:

"The analysis highlighted three hemizygous tandem repeat (TR) expansion outliers in male probands, each inherited from the mother (Supplementary Fig. 9): **the intron of IL1RAPL1 (Interleukin 1 Receptor Accessory Protein Like 1, SFARI score 2 gene), the intron of F9 (Coagulation Factor IX), and the intergenic region between MXRA5 (Matrix Remodeling Associated 5) and SNORA48B (Small Nucleolar RNA, H/ACA Box 48B)**. These longer TR noncoding variants have only been observed in heterozygous states in females, and variants of such lengths have yet to be observed in controls. **Notably, the corresponding probands exhibited features commonly**

associated with neurodevelopmental delay (Supplementary Fig. 9): IQ scores of 18, 13, and 20, and calibrated severity scores (CSS) of 9, 6, with the CSSdata for the third not available."

Regarding ancestry, the 13900 family has a 100% probability of AMR ancestry, while families 13700 and 14317 each have over 98% probability of EUR ancestry. These predictions were made using Somalier (v0.2.19 (Pedersen et al. 2020)) and were also reported in Supplementary Data 1.

page 10, line 285: I guess between "....SFARI score 2 gene). andBecause LRS data allow..." whould be a new paragraph

We agree with the reviewer and revised the manuscript to start a new paragraph at "Because LRS data allow...", as suggested.

Reviewer #3 (Remarks to the Author):

The genetic basis of autism remains underexplored. In this study, the authors performed genome sequencing of 189 individuals from 51 families with unsolved cases and generated a set of high-quality assemblies. These assemblies were compared with publicly available high-quality assemblies from healthy donors, leading to the discovery of three pathogenic variants in *TBL1XR1*, *MECP2*, and *SYNGAP1*, as well as nine candidate variants, most of which were missed by short-read sequencing. The study is well designed and solid. I have a few concerns as outlined below.

Major comments

MECP2 is structurally complex with diverse SVs and CNV events. I am wondering whether this case also harbors any other events besides the discovered deletion. Moreover, what is the actual deleted sequence in the last exon region, and what is the precise effect of this deletion on the last exon?

Because we have nearly complete sequence data, we revisited this patient's genome and although there are 490 rare SVs that are not observed in controls, none are plausible candidates for Rett syndrome (Supplementary Data 4). The deletion we discovered disrupts the open-reading frame of *MECP2*, leading to a premature stop codon that truncates the protein by approximately 140 amino acids (as shown below). Similar deletions involving exon 4 have been reported in Rett patients (Hardwick et al. 2007, Poleg et al. 2025); after consultation with our colleague Dr. Zoghbi, we are quite confident that this is the causative variant for this patient.

We revised the text to:

“Two additional pathogenic variants were discovered by LRS among the five autism-diagnosed females with features reminiscent of Rett syndrome. This included an 874 bp *de novo* DEL in *MECP2* in HYZ207_p1, which effectively disrupts the last exon of the gene **and introduces a premature stop codon, truncating the protein by approximately 140 amino acids** (Fig. 4b). **Similar deletions involving exon 4 have been reported in Rett patients (Hardwick et al. 2007, Poleg et al. 2025).** This pathogenic variant was previously missed in three rounds of clinical testing, including two gene panel sequencing tests through ARUP Laboratories and Quest Diagnostics, and one test of WES through Ambry Genetics. It was confidently identified in our LRS analysis and subsequently validated using all three sequencing platforms.”

CLUSTAL O(1.2.4) multiple sequence alignment

```

NP_004983.1      MVAGMLGLREEKSEDQDLQGLKDKPLKFKKVKKDKKKEEKEGKHEPVQPSAHHSAEPAEAG      60
HZ2207_p1       MVAGMLGLREEKSEDQDLQGLKDKPLKFKKVKKDKKKEEKEGKHEPVQPSAHHSAEPAEAG      60
*****

NP_004983.1      KAETSEGSAPSAPVPEASAPKQRRSIIRDRGPMYDDPTLPEGWTRKQKRSGRSAGKY      120
HZ2207_p1       KAETSEGSAPSAPVPEASAPKQRRSIIRDRGPMYDDPTLPEGWTRKQKRSGRSAGKY      120
*****

NP_004983.1      DVYLINPQGKAFRSKVELIAYFEKVGDTSLDPNDFDFTVTGRGSPSRREQPPKPKSPK      180
HZ2207_p1       DVYLINPQGKAFRSKVELIAYFEKVGDTSLDPNDFDFTVTGRGSPSRREQPPKPKSPK      180
*****

NP_004983.1      APGTGRGRGRPKGSGTTRPKAATSEGVQVKRVLEKSPGKLLVKMPFQTSPPGKAEGGGAT      240
HZ2207_p1       APGTGRGRGRPKGSGTTRPKAATSEGVQVKRVLEKSPGKLLVKMPFQTSPPGKAEGGGAT      240
*****

NP_004983.1      TSTQVMVIKRPGRKRKAEADPQAIPKKRGRKPGSVVAAAAAEAKKAVKKESSIRSVQETV      300
HZ2207_p1       TSTQVMVIKRPGRKRKAEADPQAIPKKRGRKPGSVVAAAAAEAKKAVKKESSIRSVQETV      300
*****

NP_004983.1      LPIKKRKTRETVSIEVKEVVKPLLSTLGEKSGKGLKTKCSPGRKSKESSPKGRSSASS      360
HZ2207_p1       LPIKKRKTRETVSIEVKEVVKPLLSTLGEKSGKGLKTKCSPGRKSYVPRWDREV-----      355
*****

NP_004983.1      PPKKEHHHHHHHSESPKAPVPLPPLPPPPPEPESEEDPTSPPEPQDLSSSVCKEEKMPR      420
HZ2207_p1       -----                                  355

NP_004983.1      GGSLESDGCPKEPAKTQPAVATAATAAEKYKHRGEGERKDIVSSSMRPNREEPVDSRTP      480
HZ2207_p1       -----                                  355

NP_004983.1      VTERVS 486
HZ2207_p1       ----- 355

```

The paper includes a large control cohort, which is very good. However, controls appear underrepresented for EUR ancestry (e.g., 39 EUR among 569). Please report the ancestry and sex composition of controls vs. families, and perform ancestry- and sex-matched analyses.

The reviewer is correct—the pangenome has generally been biased against Europeans in favor of samples of African origin in order to maximize diversity. In Supplementary Data 1.2, there are a total of 73 Europeans in the full set of 569 controls. For each control, we indicated which nested control sets it belongs to, with the smallest set listed first. For example, a sample present in all three sets is labeled “**108controls, 285controls, 569controls**” while a sample in the medium and largest sets is labeled “**108controls, 285controls**” and a sample present only in the largest set is labeled “**569controls.**” This format allows readers to easily track the inclusion of each sample across the nested control sets. We added a detailed breakdown of ancestry and sex for

both controls and autism families as part of our supplement (Supplementary Data 1.3), as shown below.

Ancestry	F	M	total
108controls	60	48	108
AFR	30	22	52
AMR	14	10	24
EAS	8	7	15
EUR	3	5	8
SAS	5	4	9
285controls	144	141	285
AFR	56	39	95
AMR	28	24	52
EAS	21	38	59
EUR	19	15	34
SAS	20	25	45
569controls	320	249	569
AFR	115	92	207
AMR	48	31	79
EAS	51	52	103
EUR	43	30	73
SAS	63	44	107
189samples	107	82	189
AFR	2	3	5
AMR	12	12	24
EAS	4	1	5
EUR	87	64	151
SAS	2	2	4

87children	56	31	87
AFR	1	1	2
AMR	8	4	12
EAS	2	1	3
EUR	44	24	68
SAS	1	1	2

We have already performed a sex-matched analysis. To clarify, we moved the statement, **"For autosomal SVs, we retained heterozygous and homozygous SVs present only in the children but not in the controls. For SVs on the sex chromosomes, we performed sex-matched comparisons and filtered SVs seen in controls with the same sex"** to "3. Rare SV pool discovery."

Per the referee's request, we performed an ancestry-matched analysis where we selected 38 families of primarily European ancestry (140 individuals, ~74%) and repeated the rare SV discovery using the 73 European controls. From the largest merged set with 569 controls, we extracted 177,522 nonredundant SVs (~40% of the total 445,142 SVs). The ancestry- and sex-matched analysis resulted in an average of 754 rare SVs per child and ~74% of which were filtered out by diverse populations. We incorporated both of the reviewer's suggestions into the same paragraph and added the requested analysis, with the revised text underlined to indicate changes made in response to Reviewer #1:

"Increasing the number of controls, especially samples of African origin, nearly doubles the number of SVs from 271,375 (108 controls) to 445,142 (569 controls) nonredundant SVs (Supplementary Data 5-8). We generated a population-level SV reference and compared the MAF distribution between the 108 controls and the largest dataset comprising 569 controls and 102 unrelated parents (n = 1,342). Based on this comparison, we estimated the fraction of variants captured at different MAF thresholds (Supplementary Fig. 12). Notably, only 16.2% of SVs with MAF < 0.1% in the larger dataset were detected in the 108-control set, illustrating the limited sensitivity of the smaller reference for identifying ultra-rare variants. **In an ancestry- and sex-matched analysis restricted to individuals of European ancestry (38 autism families and 73 controls), each child carried an average of 754 rare SVs.** Focusing on rare variants absent from the largest 569-control set, the number of rare SVs concomitantly drops to 202 events per sample (Fig. 5b), **corresponding to ~74% of rare variants being filtered out by the more diverse population controls. This demonstrates the**

substantial gain in sensitivity for removing common and low-frequency variants when using a larger, heterogeneous reference panel. As a result, 99% of common SVs are excluded per individual forming a more tractable and potentially biomedically relevant set of rare variants for downstream interpretation and enrichment analyses.”

In Fig. 1, the authors showed the N50 of contigs in autism family assemblies. However, providing NGx curves against GRCh38 and CHM13 would still be valuable. Comparing family assemblies with control assemblies would also be helpful.

We added this NGx curve to the Supplementary Figure 2 and mention it in the main text:

“The resulting assemblies are highly contiguous (average contig N50 of 43 Mbp (**Fig. 1c and Supplementary Fig. 2**)) and highly accurate (QV=56).”

Supplementary Figure 2. NG(x) curves show cumulative assembly contiguity across sequencing batches. The NG(x) statistic represents the contig length at which x% of the estimated genome size is covered. Each color corresponds to a distinct sequencing batch: reference genome CHM13 (purple), GRCh38 (black), autism families (yellow), HGSVC controls (blue), HPRC controls (green), and 1KGP ONT controls (pink).

The authors claim a “modest rate of pathogenic variant discovery (5.9%).” Did the authors consider evaluating compound heterozygous and oligogenic models (e.g., small variants, or small variants + SVs)? This may help recover additional pathogenic candidates.

We did indeed consider both of these as we thought similarly. We found no evidence of compound heterozygotes and the sample size is unfortunately far too small to observe any oligogenic SV burden. Our previous analysis suggests that at least 2000 families (Wilfert et al., 2021) are required to observe a signal in aggregate for protein-disruptive mutations. All the promising anecdotal examples we identified were unfortunately observed in siblings.

Please describe unpublished tools (e.g., subseq, BoostSV) and provide versions, key parameters, and workflow availability to enhance reproducibility.

The GitHub documents of subseq (<https://github.com/EichlerLab/subseq-smk>, Ebert et al. 2021) and BoostSV (v1.0, <https://github.com/jiadong324/BoostSV>) have been updated with more details. The overall workflow, including all the steps and tools used in this study, can be found on GitHub: <https://github.com/EichlerLab/asap>.

To make the parameters and version clearer, we added them to the Methods:

“2. Callable region evaluation using BoostSV (v1.0). To ensure the SVs fall within confidently callable regions across samples in a single family, we developed a tool, BoostSV (<https://github.com/jiadong324/BoostSV>), leveraging a machine-learning approach trained on control samples (Porubsky, Dashnow, et al. 2025). This tool evaluates read support, mapping quality, and data quality metrics from alignments surrounding the target SVs in each parent. A quality threshold of 0.5 was applied to obtain the transmission.

...

5. Read-based support validation using subseq. To further assess SV transmission, subseq (<https://github.com/EichlerLab/subseq-smk>, (Ebert et al. 2021)) was used to quantify read support for each SV in parental genomes. A dynamic window size was determined based on the SV size, and the number of reads traversing the window were counted. **The `{size50_1_1}` parameter was used, requiring a minimum of one read supporting the SV while allowing a 50% size deviation penalty.**”

Minor comments

Fig. 2a: Distinguish proband vs. sibling more clearly (shapes or colors).

The colors distinguishing probands and siblings have been updated in Fig. 2a to improve clarity.

Fig. 4c,d: Correct “Genehance” → “GeneHancer.”

The label has been corrected from “Genehance” to “GeneHancer” in Fig. 4c,d.

Typo: “TBXLR1” → “TBL1XR1.”

The text has been corrected to TBL1XR1. Thank you for catching this.

I would like to thank the authors for their contribution to the field through this work.

We greatly appreciate the reviewer’s detailed and helpful comments.

Reviewer #3 (Remarks on code availability):

I would appreciate for providing the documents for some unpublished tools.

Unpublished tools with descriptions have been added to GitHub.

RESPONSE TO REVIEWERS' COMMENTS

Reviewer #1 (Remarks to the Author):

The authors have thoroughly addressed all of my comments. The manuscript has improved substantially and is an important contribution to the field.

We thank the reviewer for their time and efforts and recognizing the improvements to the manuscript.

Reviewer #1 (Remarks on code availability):

The code and documentation for the analysis is very detailed and the authors have done a great job.

I noticed many links say "Internal path" and link to an Eichler Lab site which requires login info, and therefore is not publicly accessible or fully reproducible.

We removed any internal use paths/links in the GitHub (<https://github.com/EichlerLab/asap>) and all code is now publicly accessible to allow for full reproducibility.

Reviewer #3 (Remarks to the Author):

I have carefully reviewed the authors' revisions, and I think all of my previous concerns have been fully addressed. I have no further comments at this time.

We thank the reviewer for their time and efforts reviewing this manuscript.

Reviewer #3 (Remarks on code availability):

Source code has been properly deposited in several public repositories. I thank the authors for their efforts.

We confirm code is available on GitHub (<https://github.com/EichlerLab/asap>).